# Self-organized hetero-nanodomains actuating super Li⁺ conduction in glass ceramics

Yantao Wang[1,2,9], Hongtao Qu [3,9], Bowen Liu[4,9], Xiaoju Li[5,9], Jiangwei Ju[1] ✉, Jiedong Li[1], Shu Zhang[1], Jun Ma [1], Chao Li [4], Zhiwei Hu[6], Chung-Kai Chang[7], Hwo-Shuenn Sheu [7], Longfei Cui[1], Feng Jiang[1], Ernst R. H. van Eck[3], Arno P. M. Kentgens [3] ✉, Guanglei Cui [1,2] ✉ & Liquan Chen[8]

Easy-to-manufacture Li₂S-P₂S₅ glass ceramics are the key to large-scale all-solid-state lithium batteries from an industrial point of view, while their commercialization is greatly hampered by the low room temperature Li⁺ conductivity, especially due to the lack of solutions. Herein, we propose a nanocrystallization strategy to fabricate super Li⁺-conductive glass ceramics. Through regulating the nucleation energy, the crystallites within glass ceramics can self-organize into hetero-nanodomains during the solid-state reaction. Cryogenic transmission electron microscope and electron holography directly demonstrate the numerous closely spaced grain boundaries with enriched charge carriers, which actuate superior Li⁺-conduction as confirmed by variable-temperature solid-state nuclear magnetic resonance. Glass ceramics with a record Li⁺ conductivity of 13.2 mS cm⁻¹ are prepared. The high Li⁺ conductivity ensures stable operation of a 220 μm thick LiNi₀.₆Mn₀.₂Co₀.₂O₂ composite cathode (8 mAh cm⁻²), with which the all-solid-state lithium battery reaches a high energy density of 420 Wh kg⁻¹ by cell mass and 834 Wh L⁻¹ by cell volume at room temperature. These findings bring about powerful new degrees of freedom for engineering super ionic conductors.

Sulfide electrolytes have attracted much attention among the all-solid-state battery (ASSB) community in terms of their favorable formability and high Li⁺ conductivity ($\sigma_{Li^+}$)[1]. Among various kinds of sulfides, Li₂S-P₂S₅ glass ceramics (GCs) are among the first introduced already in 1980[2]. They remain very promising candidates for the simple element composition and mild preparation conditions in comparison to the recently developed thiophosphate Li₁₀GeP₂S₁₂ or argyrodite Li₆PS₅Cl[3,4]. Yet, the GCs' $\sigma_{Li^+}$ typically lower than 1 mS cm⁻¹ at room temperature, greatly hampers their application in ASSBs. To overcome this problem, various attempts have been made to substitute either cations such as Ce⁴⁺, Sn⁴⁺, and Sb⁵⁺ at the P-site, or anions like O²⁻ and Cl⁻ at the S-site[5–8]. However, on the one hand, the complicated element composition resulting from the multi-element doping greatly increases the preparation difficulties to obtain the target products with high

¹Qingdao Industrial Energy Storage Research Institute, Qingdao Institute of Bioenergy and Bioprocess Technology, Chinese Academy of Sciences, Qingdao 266101, PR China. ²School of Future Technology, University of Chinese Academy of Sciences, Beijing 100049, PR China. ³Institute for Molecules and Materials, Radboud University, 6525 AJ Nijmegen, The Netherlands. ⁴School of Materials Science and Engineering, Tianjin University of Technology, Tianjin 300384, PR China. ⁵State Key Laboratory of Microbial Technology, Shandong University, Qingdao 266237, PR China. ⁶Max Plank Institute for Chemical Physics of Solids, Nothnitzer Strasse 40, D-01187 Dresden, Germany. ⁷National Synchrotron Radiation Research Center, Hsinchu, Taiwan 30076, Republic of China. ⁸Beijing National Laboratory for Condensed Matter Physics, Institute of Physics, Chinese Academy of Sciences, Beijing 100190, PR China. ⁹These authors contributed equally: Yantao Wang, Hongtao Qu, Bowen Liu, Xiaoju Li. ✉e-mail: jujw@qibebt.ac.cn; a.kentgens@nmr.ru.nl; cuigl@qibebt.ac.cn

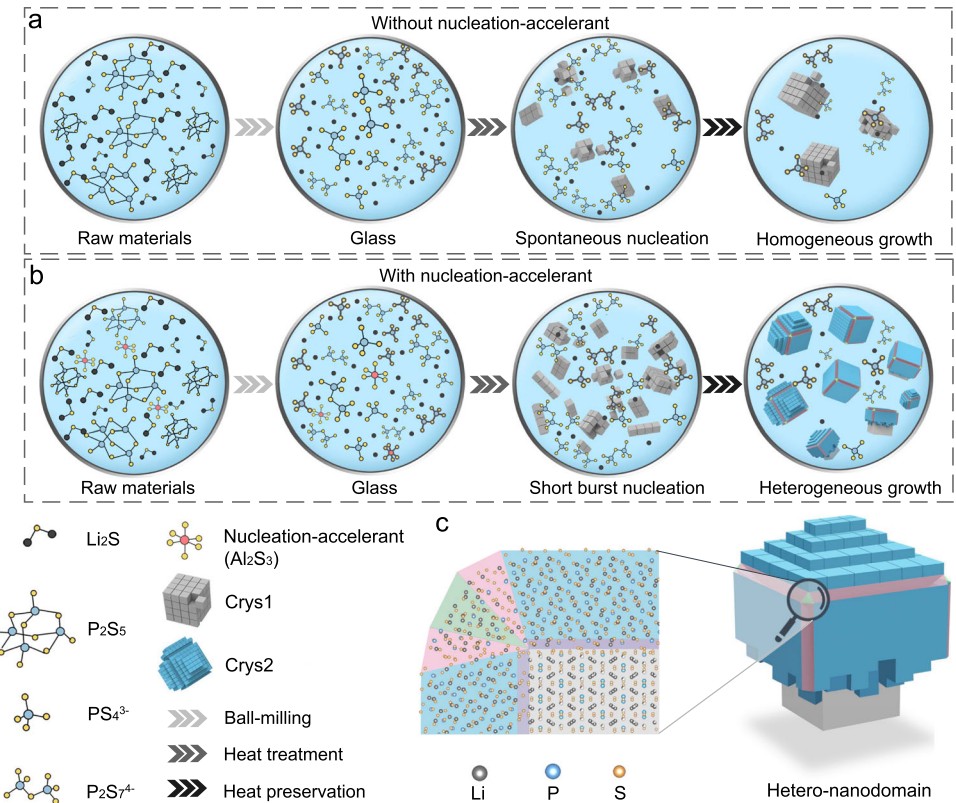

**Fig. 1 | Schematic illustration of the sequential crystallization and heterogeneous growth model (mechanism).** GC's crystallization (**a**) without and (**b**) with nucleation-accelerant. **c** A schematic diagram of 3-dimentional and 2-dimentional self-organized hetero-nanodomain. Light blue area, Crys2 shell nanocrystallites; Grey area, Crys1 core nanocrystallite; Red, violet, and green area, grain boundaries.

purity. Moreover, these heavy doping elements, undoubtedly increase the electrolyte mass, thereby lowering the specific energy. What's worse, the valence-variable elements, e.g., $Sb^{5+}$, $Ce^{4+}$, can be easily reduced to generate highly resistive phases upon contact with low-voltage anodes.

In 2013, Liu et al. developed a highly conductive GC by a solvent-assisted method. Analysis attributed the anomalous conduction phenomenon to a microstructure-mediated surface conduction mechanism[9]. Although the wet-chemical method is less suitable for large-scale synthesis than the solid-state reaction due to solvent volatilization and toxicity[10], it is favored to pursue the desirable $\sigma_{Li^+}$ through (micro)structure tuning. In reality, the GCs' $\sigma_{Li^+}$ depends highly on the crystallites' microstructural characteristics like grain size, morphology or distribution, etc[11]. As Maier points out, reducing the crystallite size to the nanoscale has a crucial impact on mass transfer[12]. Nanostructured materials differ from their conventional bulk counterparts by a large fraction of grain boundaries, which are so closely spaced that charge carrier channels for fast migration exist[13]. As a classic example, in the nanostructured hetero-domain (hetero-nano-domain) $AgI/Al_2O_3$[14], the $Ag^+$ conductivity is considerably improved due to the more abundant charge carriers and lower diffusion energy barrier along the grain boundaries compared to the bulk[15]. These scenarios motivate us to present a strategy for nanocrystallization to significantly increase the grain boundary densities and thereby to optimize the GCs' $\sigma_{Li^+}$, which, to the best of our knowledge, has not yet been proposed.

Advanced techniques for obtaining nanocrystals, including molecular beam epitaxy, pulsed laser deposition, and sputtering etc., are not suitable to scale-up production. From an industrial point of view, it is extremely urgent to open a universal way to incorporate nanocrystallites in GCs via solid-state reactions. Inspired by the classic LaMer model in colloid chemistry[16], we propose here a nucleation-accelerant induced sequential crystallization and heterogeneous growth mechanism: during the solid-state reaction, the nucleation-accelerant reduces the crystal nucleation barrier to induce a short burst of nucleation forming the core nano-grains, on whose crystal faces another crystal subsequently crystallizes and self-organizes into hetero-nanodomains with a large proportion of grain boundaries.

The universality of this mechanism is confirmed through advanced characterization techniques, including synchrotron X-ray diffraction (SXRD), static and magic angle spinning solid-state nuclear magnetic resonance (MAS SSNMR), cryogenic transmission electron microscope (cryo-TEM) and electron holography. Most importantly, the resultant hetero-nanodomains can actuate superior $Li^+$-conduction along the grain boundaries, as confirmed by density functional theory (DFT) calculations and variable-temperature SSNMR. Consequently, the highest room temperature $\sigma_{Li^+}$ of 13.2 mS cm$^{-1}$ among the previously reported GCs is obtained. Such a high $\sigma_{Li^+}$ enables the successful implementation of GCs within a 220 μm-thick composite cathode (44.6 mg cm$^{-2}$ $LiNi_{0.6}Mn_{0.2}Co_{0.2}O_2$, NCM622), endowing the resulting ASSB with an excellent energy density of 420 Wh kg$^{-1}$ (by cell mass) and 834 Wh L$^{-1}$ (by cell volume) at room temperature. This approach provides a general framework and universal strategy to design super ionic conductors, which makes the present findings highly relevant for the development of ASSBs.

## Results and discussion
### Sequential crystallization and heterogeneous growth model
The crystallization of glass mainly consists of nucleation and crystal growth. Without disturbances, the nucleus will grow freely to form a crystalline mass (spontaneous nucleation and homogeneous growth, Fig. 1a). If nanocrystallites are desired, the crystallization must be interrupted. Based on the LaMer model, two essential prerequisites must be met to achieve nanocrystallization: short burst nucleation and

**Table 1 | Molar ratio of the reagents (Li₂S, P₂S₅, Al₂S₃) in Al-GCs and the corresponding denotation**

| Li$_2$S | P$_2$S$_5$ | Al$_2$S$_3$ | Ratio of (Li$_2$S: P$_2$S$_5$) | Denotation |
|---------|-----------|-------------|-------------------------------|------------|
| 56.25 | 18.75 | 0 | 75:25 | LPS7525 |
| 55.5 | 19.5 | 1 | 74:26 | LPS7426 |
| 54.75 | 20.25 | 2 | 73:27 | LPS7327 |
| 54 | 21 | 3 | 72:28 | LPS7228 |
| 53.25 | 21.75 | 4 | 71:29 | LPS7129 |
| 52.5 | 22.5 | 5 | 70:30 | LPS7030 |
| 51.75 | 23.25 | 6 | 69:31 | LPS6931 |
| 48.75 | 26.25 | 10 | 65:35 | LPS6535 |

crystal growth suppression, in which the nucleation-accelerants play an indispensable role[17,18]. Additionally, the solid-state chemistry of the Li$_2$S-P$_2$S$_5$ binary system indicates that the crystalline substances depend on the fraction of Li$_2$S or P$_2$S$_5$, i.e., the P/S ratio[19]. Accordingly, the nucleation-accelerants should have the following functions to induce sequential crystallization and heterogeneous growth to form hetero-nanodomains: 1) substantially lowering the nucleation barrier of crystallite 1 (Crys1) and thus triggering its burst nucleation (Fig. 1b); 2) changing the P/S ratio around the as-formed nuclei to suppress their continuous growth. Therefore, the altered P/S ratio triggers the formation of different crystallite (Crys2). Based on the fact that heterogeneous nucleation is usually kinetically more favorable than homogeneous nucleation[20], Crys2 tends to nucleate and grow upon Crys1. More importantly, Crys2 growing on different crystal faces of Crys1 shows different orientations, limiting the coalescence of Crys2 to bulk. Hence, under the action of nucleation-accelerant, Crys1 and Crys2 self-organize into hetero-nanodomains.

As indicated in Fig. 1c, within the hetero-nanodomain, the difference in crystal structure between Crys1 and Crys2 result in geometric misfits, that is, heterogeneous grain boundaries, while adjacent Crys2 with different orientations cause geometric misfits of homogeneous grain boundaries. For a perfect hetero-nanodomain, there are 12 grain boundaries (red) resulting from two adjacent Crys2 nanocrystallites, 8 grain boundaries (green) resulting from three adjacent Crys2 nanocrystallites, and 6 grain boundaries (violet) between Crys1 and Crys2 nanocrystallites. The large proportion of grain boundaries occurring in the self-organized hetero-nanodomains are predicted to significantly enhance the $\sigma_{Li^+}$ of GCs.

Guided by this model, a series of nucleation-accelerants including Al$_2$S$_3$, Ga$_2$S$_3$, and SiS$_2$ are screened to verify the universality of this mechanism. All of these nucleation-accelerants can induce the sequential crystallization and heterogeneous growth to self-organize into hetero-nanodomains in GCs, significantly improving the $\sigma_{Li^+}$ to 10 mS cm$^{-1}$ at room temperature. It is believed that, in addition to the three nucleation-accelerants provided in this work, more nucleation-accelerants will be discovered based on this universal mechanism. In the following results and discussion, we focus on Al$_2$S$_3$ as the nucleation-accelerant to validate the model (mechanism), while the results for Ga$_2$S$_3$ and SiS$_2$ can be found in the Supplementary Information.

### Species identification of Al$_2$S$_3$-tuned GCs (Al-GCs)

As indicated in Table 1, the Al-GCs are obtained by adding Al$_2$S$_3$ nucleation-accelerant to regulate the molar ratio of Li$_2$S: P$_2$S$_5$ from 75: 25 to 65: 35, the ratio ranges commonly studied[19]. The raw materials are mixed by mechanical grinding and subsequently subjected to a heat treatment. Supplementary Fig. 1a and Supplementary Fig. 1b outline the obtained Al-GCs with different phase compositions identified by SXRD and Raman spectra. Conclusively, as visualized in Supplementary Fig. 1h, at a decreasing Li$_2$S: P$_2$S$_5$ ratio from 75: 25 to 65: 35, the phase composition undergoes the crystal structure

transformation of Li$_3$PS$_4$ analog phase → Li$_7$P$_3$S$_{11}$ analog phase → Li$_4$P$_2$S$_6$ analog phase.

The subtle changes in the local structure of Al-GCs are also detected by $^{31}$P MAS NMR spectra. As shown in Fig. 2a, for both LPS7525 and LPS7426, one single resonance at 87 ppm with a small shoulder at 89 ppm is detected, indicating the presence of isolated PS$_4^{3-}$ polyhedra. For LPS7327, in addition to the 89 ppm peak, another peak at 91 ppm is observed simultaneously. The resonances at 91 ppm can be attributed to P$_2$S$_7^{4-}$[21]. Further decreasing Li$_2$S, P$_2$S$_6^{4-}$ units (104 and 108 ppm) are detected and continuously increasing with decreasing Li$_2$S content, which is in good accordance with the SXRD and Raman results. To further corroborate the bond connectivity in the P − S units, a 2D $^{31}$P − $^{31}$P INADEQUATE NMR experiment was performed for LPS7228. The 2D experiment is based on *J*-coupling which allows us to probe P − P and/or P − S − P bond connectivity. Figure 2b clearly shows that an intense peak located at 91 ppm attributed to the P − S − P bond in P$_2$S$_7^{4-}$ is observed. Interestingly, there are three peaks at (104, 208), (105, 213) and (108, 213) stemming from the P − P bond in P$_2$S$_6^{4-}$. The peak on the diagonal arises from the P − P correlation between two identical P sites, while the width of peaks parallel to the diagonal hints at the slightly different P environment in P$_2$S$_6^{4-}$ units. This could be explained by the disordered conformation of [P$_{4/2}$S$_6$]$^{4-}$ anion in which only two out of four possible P sites are simultaneously occupied[22].

A more detailed characterization of the evolution of species in Al-GCs based on $^{27}$Al and $^6$Li SSNMR spectra can be found in a separate publication[23], this suggests that Al is first introduced to partially occupy the vacant LiS$_6$ sites in Li$_3$PS$_4$ to form Li$_{3-3x}$Al$_x$PS$_4$, and the solid solubility is 0.06. Beyond the solid solubility limit, lithium-depleted thiophosphates, such as Li$_7$P$_3$S$_{11}$ and Li$_4$P$_2$S$_6$, begin to form with decreasing Li$_2$S content. The evolution of species in Al-GCs is visualized in Fig. 2c.

### Hetero-nanodomains

The solid chemistry of the Li$_2$S-P$_2$S$_5$ binary system indicates that the crystalline substance depends strongly on the P/S ratio in the starting materials (Supplementary Fig. 4a). For example, when the molar ratio of Li$_2$S to P$_2$S$_5$ is 75: 25 (P/S = 0.25), pure crystalline Li$_3$PS$_4$ can be obtained, with all sulfur atoms are in PS$_4^{3-}$ tetrahedra. Upon decreasing the ratio to 70: 30 (P/S = 0.273), a pure Li$_7$P$_3$S$_{11}$ phase is deposited consisting of PS$_4^{3-}$ isolated and corner-sharing P$_2$S$_7^{4-}$ tetrahedra with a 1: 1 molar ratio. Decreasing the Li$_2$S fraction further to 67 mol% (P/S = 0.333), pure Li$_4$P$_2$S$_6$ phase crystallizes, yielding P$_2$S$_6^{4-}$ dumbbells.

Because both P$_2$S$_7^{4-}$ and P$_2$S$_6^{4-}$ derive from PS$_4^{3-}$ (Supplementary Fig. 4b), in this work, the P/S ratio of all sample reagents are fixed to 0.25, where the P source is P$_2$S$_5$ and the S sources are Li$_2$S, P$_2$S$_5$ and Al$_2$S$_3$. Both SXRD and $^{27}$Al MAS SSNMR results suggest that Al$_2$S$_3$ can participate in the solid-state reaction to form Al-doped Li$_3$PS$_4$, i.e., Li$_{3-3x}$Al$_x$PS$_4$, while the $^{27}$Al MAS SSNMR results further indicate that the maximum of the Al doping content, i.e., the solubility limit of Al, is only 0.06. Below this threshold, all the Al$_2$S$_3$ is reactive and thus the P/S ratio is constant 0.25 to generate Li$_{3-3x}$Al$_x$PS$_4$. Beyond the solubility limit, the S source from the residual Al$_2$S$_3$ is nonreactive and as a result the P/S ratio is higher than 0.25. In these cases, Li$_{2.82}$Al$_{0.06}$PS$_4$ (P/S = 0.25), Li$_7$P$_3$S$_{11}$ (P/S = 0.273) and Li$_4$P$_2$S$_6$ (P/S = 0.333) crystallize sequentially based on the $\Delta G_r$ (Gibbs free energy change) difference: $\Delta G_{r(Li3-3xAlxPS4)}$ (−138.38 meV atom$^{-1}$) < $\Delta G_{r(Li7P3S11)}$ (−87.27 meV atom$^{-1}$) < $\Delta G_{r(Li4P2S6)}$ (12.81 meV atom$^{-1}$).

Take LPS7228 for example: 72 mol% Li$_2$S, 4 mol% Al$_2$S$_3$, and 28 mol% P$_2$S$_5$ first react to form 27.1 wt % Li$_{2.82}$Al$_{0.06}$PS$_4$ due to the lowest $\Delta G_r$, which indicates Al$_2$S$_3$ substantially lowers the nucleation barrier of Li$_{2.82}$Al$_{0.06}$PS$_4$ and thus triggers its burst nucleation. By theoretically excluding the unreacted Al$_2$S$_3$, the remaining Li$_2$S and P$_2$S$_5$ lead to a P/S ratio of 0.267, just satisfying the formation condition for forming Li$_7$P$_3$S$_{11}$ (P/S ratio, 0.273). Details of the calculation can be found in the

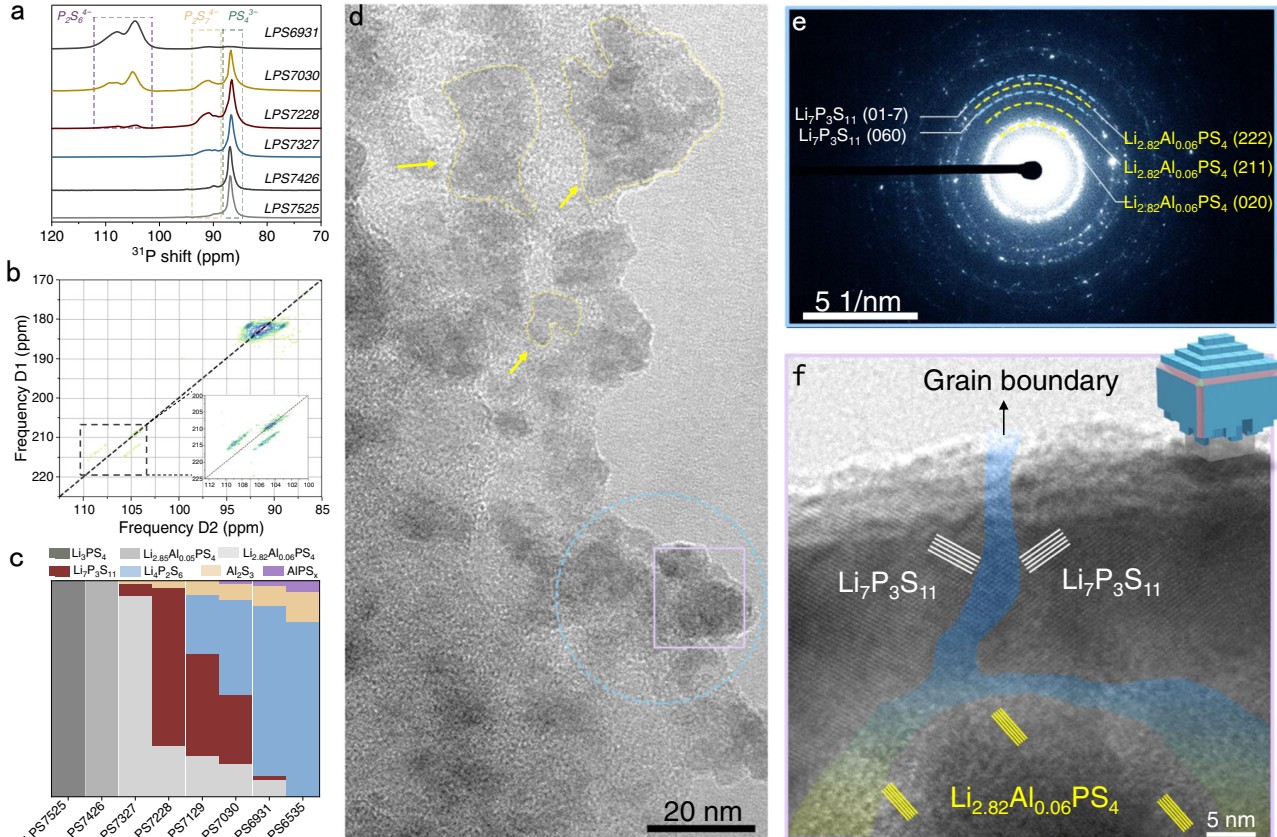

**Fig. 2 | Specific species of Al-GCs and the microstructure of hetero-nanodomains. a** $^{31}P$ MAS SSNMR spectra of Al-GCs. **b** 2D $^{31}P$ refocused INADE-QUATE spectrum of LPS7228. The contour peaks in the 2D spectrum indicate the phosphorous units that are chemically bonded. **c** Bar graph visualizing the species evolution of Al-GCs with various $Li_2S$: $P_2S_5$. **d** cryo-TEM, **e** SAED, and **f** HRTEM images of hetero-nanodomains in LPS7228.

Supplementary Information. Hence, another function of the nucleation-accelerant is demonstrated, that is, changing the P/S ratio of the as-formed $Li_{2.82}Al_{0.06}PS_4$ crystallite to suppress its continuous growth, while the changed P/S ratio triggers the formation of $Li_7P_3S_{11}$. This sequential crystallization phenomenon induced by $Al_2S_3$ is experimentally confirmed via in situ temperature-dependent XRD analysis. As shown in Supplementary Fig. 5, $Li_{2.82}Al_{0.06}PS_4$ starts to crystallize once the furnace is heated to 300 °C while $Li_7P_3S_{11}$ crystallizes only after being held at 300 °C for 40 min. Following that, the XRD spectra show that the intensity of all peaks steadily rises with heat treatment duration, implying that the degree of crystallinity increases continuously. Meanwhile, no other impurities emerge during annealing.

The hetero-nanodomains resulting from the sequential crystallization and heterogeneous growth are then investigated by cryo-TEM. As shown in Fig. 2d, a large amount of nanodomains (the dark area) with a size of 20–40 nm are observed, indicating the burst nucleation. To further characterize these nanodomains, selected area electron diffraction (SAED) is first performed for the region marked by an azury circle in Fig. 2d. The SAED patterns displayed in Fig. 2e exhibit Debye-Scherrer rings with a halo pattern, indicating that LPS7228 contains both $Li_{2.82}Al_{0.06}PS_4$ and $Li_7P_3S_{11}$ crystallites, agreeing well with the SXRD results. High resolution TEM (HRTEM) of the pink square within the SAED area is displayed in Fig. 2f, the HRTEM image reveals clear lattices with interplanar distances of 0.332 nm (marked yellow) and 0.368 nm (marked white), matching well with the $d_{121}$ and $d_{1\bar{3}0}$ spacing of $Li_{2.82}Al_{0.06}PS_4$ and $Li_7P_3S_{11}$, respectively[24,25]. Interestingly, this image reveals that the $Li_{2.82}Al_{0.06}PS_4$ nanocrystallites are closely surrounded by $Li_7P_3S_{11}$

nanocrystallites to form $Li_7P_3S_{11}@Li_{2.82}Al_{0.06}PS_4$ hetero-nanodomains. DFT calculations also suggest the preferential nucleation and growth of $Li_7P_3S_{11}$ crystallites on $Li_{2.82}Al_{0.06}PS_4$ crystal faces (Supplementary Fig. 6). In such hetero-nanodomains, the proportion of 5 nm-wide grain boundaries resulting from the geometric mismatch between the $Li_{2.82}Al_{0.06}PS_4$ core nanocrystallites and the $Li_7P_3S_{11}$ shell nanocrystallites, or among the adjacent $Li_7P_3S_{11}$ nanocrystallites with different orientations, increases substantially.

## Li$^+$-conduction

To gain in-depth insight into the correlation between microstructure and conductivity, the trend in $\sigma_{Li^+}$ as determined from electrochemical impedance spectroscopy (EIS) is plotted. According to Fig. 3a and Supplementary Fig. 7, a value of 0.20 mS cm$^{-1}$ is reached for LPS7525 at room temperature and increases to 0.64 mS cm$^{-1}$ for LPS7426. With decreasing $Li_2S$: $P_2S_5$ ratio, $\sigma_{Li^+}$ increases to 3.3 mS cm$^{-1}$ for LPS7327, and even reaches a maximum of 13.2 mS cm$^{-1}$ for LPS7228. Further reduction in the $Li_2S$: $P_2S_5$ ratio reduces $\sigma_{Li^+}$ to 1.43 (LPS7129), 0.33 (LPS7030), 0.03 (LPS6931), and 0.014 mS cm$^{-1}$ (LPS6535). Supplementary Table 13 lists the $\sigma_{Li^+}$ for GCs prepared using the solid-state reaction method reported in the literature. Clearly, sample LPS7228 studied in this work reaches the highest value reported so far.

SXRD analysis suggests that LPS7228 consists of $Li_{2.82}Al_{0.06}PS_4$ (27.1 wt%) and $Li_7P_3S_{11}$ (72.9 wt%) with a small quantity of $Al_2S_3$ residues, Fig. 2c. According to the EIS results in Supplementary Fig. 10, $\sigma_{Li^+}$ of the as-prepared $Li_{2.82}Al_{0.06}PS_4$ and $Li_7P_3S_{11}$ is estimated to be 0.94 and 1.4 mS cm$^{-1}$, respectively. In addition, the factual composite of 27.1 wt% $Li_{2.82}Al_{0.06}PS_4$ and 72.9 wt% $Li_7P_3S_{11}$ exhibits a $\sigma_{Li^+}$ of 1.01 mS cm$^{-1}$. All are much lower than the value of 13.2 mS cm$^{-1}$ for LPS7228.

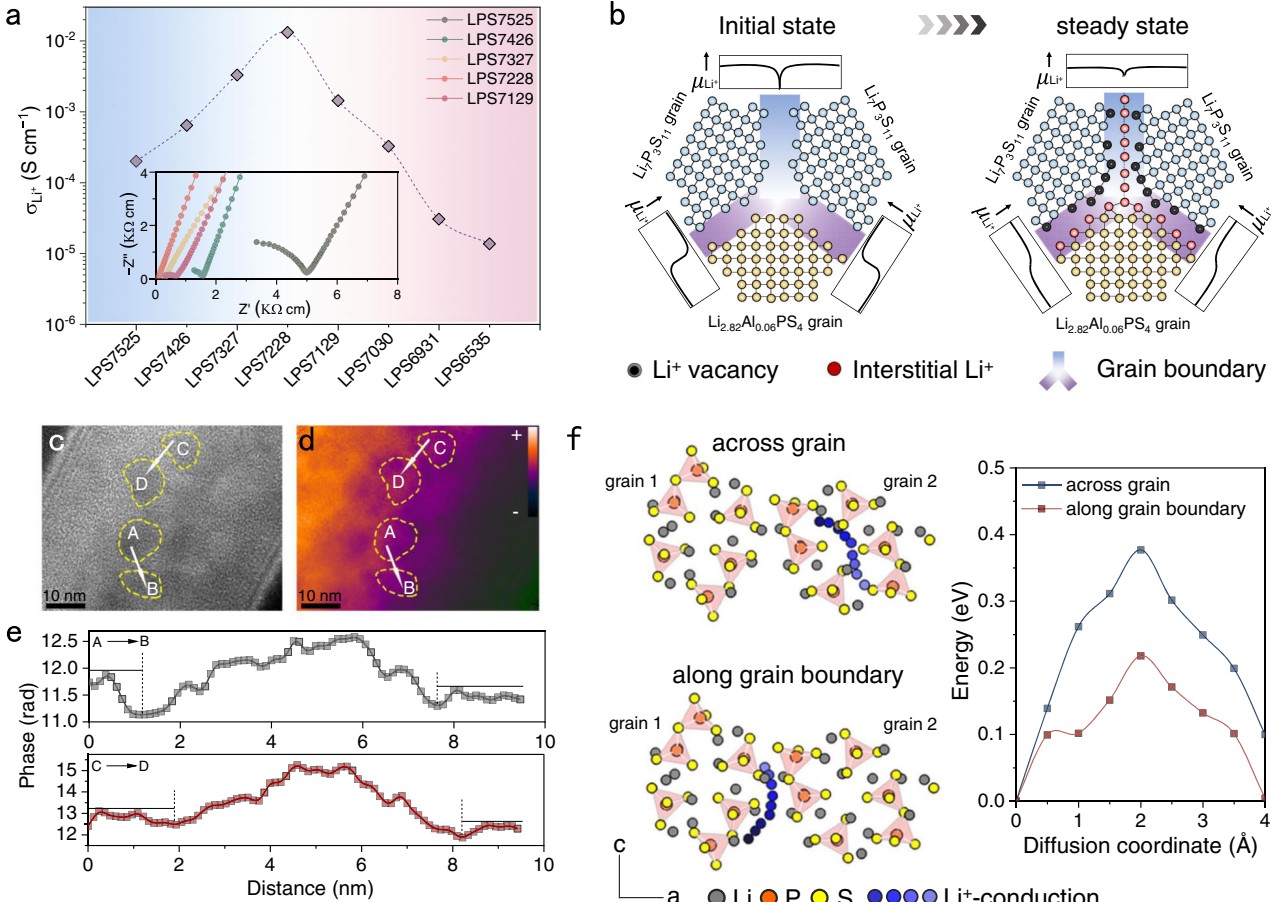

**Fig. 3 | The super Li+-conduction. a** $\sigma_{Li^+}$ at room temperature of various Al-GCs in the cold-pressed state. EIS plots are shown in the inset. **b** Schematic diagram of the carrier redistribution within $Li_7P_3S_{11}@Li_{2.82}Al_{0.06}PS_4$ hetero-nanodomain driven by $\mu_{Li^+}$. **c**–**e** Electron holography observation of LPS7228: **c** Initial hologram image. **d** Reconstructed phase image. Inset color bar is defined as the relative magnitude of the radius. **e** Phase profiles along the white arrow from A to B and C to D. **f** Schematic diagram of Li+ migration along grain boundary and across grain with the corresponding diffusion energy (activation energy).

Based on this observation, the high value of $\sigma_{Li^+}$ for LPS7228 is believed to originate from the make-up of the hetero-nanodomains.

As recently suggested[26], when two phases come into contact, ions (carriers) will redistribute along the grain boundaries because of the chemical potential ($\mu$) difference to form defect-rich regions that allow the carriers to migrate readily. For the $Li_7P_3S_{11}/Li_{2.82}Al_{0.06}PS_4$ heterojunction, DFT calculations suggest that $\mu_{Li^+}$ of $Li_7P_3S_{11}$ (−3.25 eV) is higher than that of $Li_{2.82}Al_{0.06}PS_4$ (−3.35 eV). Thus, a defect-rich region in which Li+ vacancies accumulate at the edge of the grain boundaries near $Li_7P_3S_{11}$ and Li+ ions accumulate at the edge near $Li_{2.82}Al_{0.06}PS_4$, as shown in Fig. 3b. In $Li_7P_3S_{11}/Li_7P_3S_{11}$ homo-junction, $\mu_{Li^+}$ is usually lower in the grain boundary core than at the edges[27]. Hence, Li+ moves spontaneously from the grain boundary edges to the core, forming a symmetric region with abundant Li+ ions in the grain boundary core sandwiched between Li+ vacancies on both grain boundary edges.

Experimental evidence of carrier redistribution is provided by off-axis electron holography because carrier redistribution causes a built-in potential change[28]. First, an initial electron hologram is acquired using an objective spherical aberration corrected TEM. Then, the reconstructed phase image is separated. In the absence of any magnetic field, the projected phase shift of the electron wave caused by electrostatic fields within the sample can be expressed simply as:

$$\Delta\varphi(x,y) = C_E \cdot \int_{path} V_{tot}(x,y,z)dz \approx C_E \cdot t(x,y) \cdot V_{tot}(x,y) \quad (1)$$

where $z$ is the optical axis, $C_E$ is an electron-energy-dependent interaction constant with the value of 0.01009 rad $V^{-1}$ $nm^{-1}$ for 80 kV electrons, $V_{tot}(x,y)$ is the average of the potential $V_{tot}(x,y,z)$ through the specimen, $t(x,y)$ is the sample thickness at a given position. By using Eq. (1), the phase shift measured by holography can be directly related to the potential distribution of the sample[29].

As shown in Fig. 3c, a large number of hetero-nanodomains can be clearly distinguished by the dark regions, among which are the grain boundaries. Phase profiles are extracted along the white arrow (from A to B and C to D, Fig. 3e) in the reconstructed phase image (Fig. 3d). The local potential distribution is highlighted in a color-density plot. As observed from the direction of the white arrow, there is an abrupt decrease in potential at the outer edges of the two adjacent hetero-nanodomains followed by a potential increase. Correspondingly, the grain boundary edges are enriched with Li+ vacancies resulting in a low potential, while the core potential increases sharply due to the enrichment of Li+ ions. This is a very direct observation of the carrier redistribution as described by our model (Fig. 3b).

With the large fraction of carrier-enriched grain boundaries, enhanced Li+ conduction is expected, which is probed by recording static variable-temperature 7Li spectra and NMR relaxometry[23]. In particular, the relaxation time in the rotating frame $T_{1\rho}$ is employed to probe Li+ dynamics in the kHz regime that is associated with long-range Li+ diffusion. It seems that only one motional process is present in LPS7525 with activation barrier of 0.13 eV. This hardly changes for LPS7426. We observe a very flat slope in the temperature-dependent

relaxivity curve of LPS7228, meaning the activation barrier is substantially lowered. The curve shows multiple slopes, however, and no well-defined maximum, implying that there is a distribution of multiple diffusion pathways, as is expected for hetero-nanodomains consisting of several phases. Moreover, DFT simulations are performed to investigate $Li^+$ diffusion barriers (activation energy, Fig. 3f) along grain boundaries or across nano-grains. It is calculated that the activation energy for $Li^+$ diffusion along grain boundaries is 0.21 eV, about 2 times lower than that of across nano-grains, 0.37 eV. According to the Nernst-Einstein Eq. (2) and the Arrhenius Eq. (3), both the enhancement of carriers' concentration and a low activation energy lead to an excellent $\sigma_{Li^+}$ of 13.2 mS cm$^{-1}$ at room temperature.

$$\sigma_{Li^+} = \frac{N \cdot D \cdot e^2}{k_B \cdot T} \qquad (2)$$

$$D = D_0 \cdot e^{-\frac{E_a}{R \cdot T}} \qquad (3)$$

where $N$ is the number of charge carriers; $D$, the self-diffusion coefficient; $e$, the electron charge; $k_B$, the Boltzmann constant; $T$, the Kelvin temperature; $D_O$, temperature-independent diffusion coefficient; $E_a$, activation energy; $R$, the gas constant.

### Demonstration in lithium metal ASSB

Lithium metal is a promising anode material for high-energy-density ASSBs because of its high theoretical capacity (3860 mAh g$^{-1}$) and low electrochemical potential (−3.04 V vs. the standard hydrogen electrode)[30]. The electrochemical performance of lithium metal ASSB depends strongly on the compatibility between electrolyte and lithium metal. Thereby, the stability of LPS7228 against lithium metal anode is firstly evaluated in symmetric Li−Li cells at room temperature. As shown in Fig. 4a and Supplementary Fig. 16, at a current density of 0.2 mA cm$^{-2}$, the overpotential gradually increases from 15.7 to 28 mV during the first 150 h, which can be attributed to the formation of stabilized interface between lithium metal and LPS7228 as discussed below. The nearly unchanged overpotential after 150 h proves that the in situ formation of the interface is self-terminating, rendering the Li|LPS7228|Li symmetric cell stable cycling for over 1400 h with specific areal capacity of 0.5 mAh cm$^{-2}$, and over 1000 h with a specific areal capacity of 1.0 mAh cm$^{-2}$. The stability and surface capacity are adequate for high-performance lithium metal ASSBs[31–33].

To reveal the stable lithium plating/stripping behavior, a device (Fig. 4b) for operando optical microscopy observation was designed in house. For a fair comparison, the most common $Li_6PS_5Cl$ is adopted as the contrastive electrolyte. Supplementary Fig. 17 displays a flat voltage plateau of 50 mV for over 10 hrs at a current density of 0.5 mA cm$^{-2}$ for Li|LPS7228|Li, while the cell using $Li_6PS_5Cl$ survives only 28 min with a short circuit appearing afterwards. Cross-sectional snapshots, taken after 10 hrs (Fig. 4c) show an integrated LPS7228 layer without cracks or lithium dendrites inside, which is also verified by X-ray microtomography (Fig. 4e). Besides, the interface between LPS7228 and lithium metal anode is very smooth. In contrast, lithium metal granules, lithium dendrites, and spiky interface between $Li_6PS_5Cl$ and the lithium metal anode are clearly observed in Fig. 4d, f and Supplementary Fig. 18. According to X-ray photoelectron spectroscopy (XPS) analysis and DFT calculations, the stable Li polarization behavior can be attributed to the Li−Al alloy formed in situ to uniformize the $Li^+$ flux during the Li plating/stripping processes[6,34]. Detailed discussion can be found in the Supplementary Information related to Supplementary Fig. 19.

The high $\sigma_{Li^+}$ and good anodic stability of LPS7228 against lithium metal suggest its very promising potential for direct application (without a buffer layer between the electrolyte and lithium metal

anode or using a modified lithium metal anode) in lithium metal ASSBs. With a LiNbO$_3$-coated LiCoO$_2$ cathode (areal loading 8.92 mg cm$^{-2}$), lithium metal ASSBs with $Li_3PS_4$, $Li_6PS_5Cl$, $Li_{10}GeP_2S_{12}$ (LGPS) and LPS7228 electrolytes are demonstrated for a fair comparison. As shown in Supplementary Fig. 21, the LiCoO$_2$|Li ASSB using $Li_3PS_4$ is only stable for the first 3 cycles. After then, the cell shows severe overcharging related to dendrite growth during charging[35]. Similarly, the LiCoO$_2$|Li ASSB with $Li_6PS_5Cl$ survives only 15 cycles, followed by a sudden voltage-drop due to lithium dendrite growth within $Li_6PS_5Cl$ (Fig. 4d, f). Rapid capacity decay is observed for LiCoO$_2$|LGPS|Li, mainly attributed to the notorious side reactions between lithium metal and LGPS, resulting in highly resistive products[36].

By a huge contrast, a lithium metal ASSB with LPS7228 demonstrates a more stable cycling performance (Fig. 4g, h). Specifically, a reversible specific discharge capacity of 141.7 mAh g$^{-1}$ is achieved for the first cycle in the voltage range of 3.0 − 4.3 V with a high CE of 92.4%, due to the excellent stability of Li/LPS7228 interface. After 100 cycles, 90.6% capacity is retained, indicating superior interface compatibility. Remarkably, a reversible specific capacity of 104.1 mAh g$^{-1}$ is maintained after 200 cycles, showing a good retention of 73.5% with an average CE of 99.8%, suggesting highly reversible $Li^+$ intercalation/deintercalation. Compared to other reported lithium metal ASSBs with the same configuration (Supplementary Table 14), the lithium metal ASSB reported in this work performs best.

### Demonstration in high energy ASSB

To assemble high specific energy ($E_m$) ASSB, a free-standing and flexible film with a thickness of 110 μm is prepared by rubbing LPS7228 powders with polytetrafluoroethylene binder (Supplementary Fig. 24a−c), which has a high room temperature $\sigma_{Li^+}$ of 10.3 mS cm$^{-1}$. Microsilicon (μSi) is used as the anode (Supplementary Fig. 24d) due to its high specific capacity exceeding 3500 mAh g$^{-1}$ and excellent C-rate performance[37]. As shown in Supplementary Fig. 24e, the battery is constructed by a 220 μm-thick composite cathode containing 80 wt% LiNbO$_3$@NCM622 with an areal loading of 44.6 mg cm$^{-2}$, a LPS7228 film, and a 20 μm-thick μSi anode (N/P = 1.21). At a charge rate of 0.1 C (1 C = 180 mAh g$^{-1}$) and room temperature, an initial discharge capacity of 171 mAh g$^{-1}$ is obtained, corresponding to an excellent areal capacity of 7.62 mAh cm$^{-2}$ and an $E_m$ of 420 Wh kg$^{-1}$ by cell mass. Encouragingly, a high value of 6.63 mAh cm$^{-2}$ corresponding to an $E_m$ of 336 Wh kg$^{-1}$ is still obtained when increasing the C-rate to 0.2 C and cycling stably over 50 cycles with an average CE of 98.1% (Fig. 4j). As included in Fig. 4k, the Ragone plots indicate that compared with the previously reported high energy ASSBs, our battery demonstrates a much higher $E_m$ at a fixed specific power ($P_m$) or vice versa[38–47], empowering our work an important potential for future development of electric vehicle batteries with both high $E_m$ and $P_m$.

In summary, this work proposes a sequential crystallization and heterogeneous growth mechanism to drive crystallites nanocrystallization during the solid-state reaction. The resulting self-organized hetero-nanodomains, composed of a considerable number of grain boundaries, can actuate super $Li^+$-conduction, leading to a high room temperature $\sigma_{Li^+}$ of 13.2 mS cm$^{-1}$ in the cold-pressed state. Such super $Li^+$-conductive GCs ensure stable operation of thick composite cathodes (>200 μm), realizing ASSBs with high energy density of 420 Wh kg$^{-1}$ by cell mass at room temperature. These relevant results hence open a whole new avenue to pursue superionic conductors.

## Methods
### Synthesis
Reagent-grade Li$_2$S (Alfa Aesar, 99.9 %), P$_2$S$_5$ (MACKLIN, ≥ 99 %), and Al$_2$S$_3$ (Alfa Aesar, 99.9%) are used as precursors. Given the different coordination polyhedra, e.g., *pyro*-thiophosphate (P$_2$S$_7$$^{4-}$) or *hypo*-thiophosphate (P$_2$S$_6$$^{4-}$), are derived from the *ortho*-thiophosphate

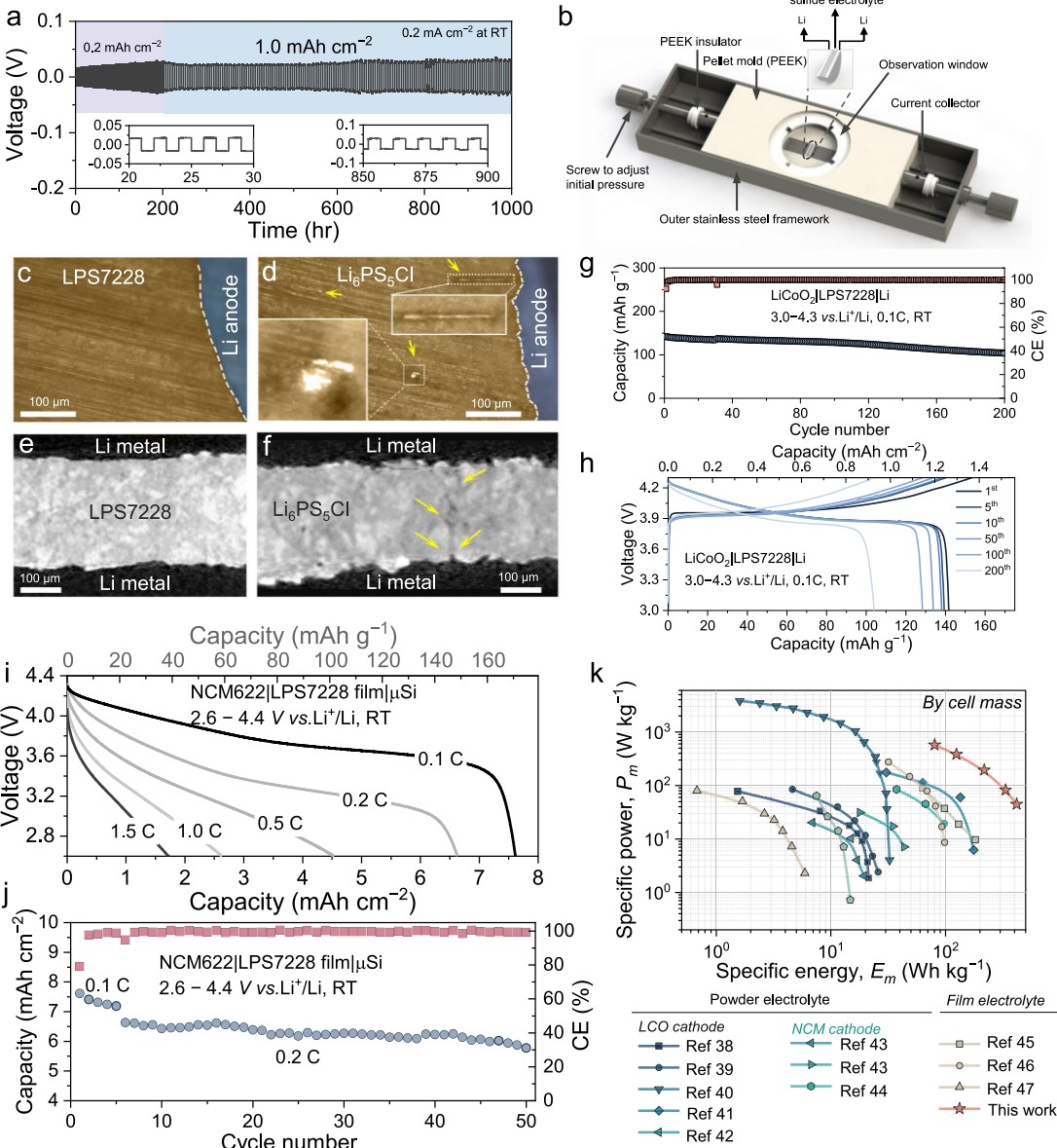

**Fig. 4 | Demonstration of LPS7228 in ASSBs. a** Galvanostatic cycling of the Li|LPS7228|Li cell at a current density of 0.2 mA cm⁻² and areal specific capacities of 0.2 or 1.0 mAh cm⁻² at room temperature. **b** Schematic view of the home-made device for operando optical microscopy observation. Top-down optical observation of (**c**) Li|LPS7228|Li and (**d**) Li|Li₆PS₅Cl|Li symmetric cells after 10 h cycling. Sliced images of the reconstructed tomographic volumes of (**e**) Li|LPS7228|Li and

(**f**) Li|Li₆PS₅Cl|Li symmetric cells after 10 h cycling. **g** Cycle behavior and (**h**) charge-discharge curves of LiCoO₂|LPS7228|Li ASSB at room temperature. **i** The C-rate performance and (**j**) cycle performance of the NCM622|LPS7228 film|µSi ASSB at room temperature. **k** Ragone plots for cells employing sulfide electrolytes cycled at room temperature. The $E_m$ and $P_m$ are delivered during discharging, normalized by cell mass.

polyhedra (PS₄³⁻)[48], the P/S ratio in various Al₂S₃-tuned GCs (Al-GCs) is fixed to 1: 4. To prepare Al-GCs, the reagents based on the stoichiometry listed in Table 1 are weighed out and mixed in zirconia jars followed by mechanical milling for 24 h at a speed of 450 rpm using an apparatus (FRITSCH, Pulverisette7). Afterwards, the obtained powders are pressed into green pellets under 320 MPa. Then the pellets are sealed into an argon-filled quartz tube and heated at 300 °C for 8 h and naturally cooled down to room temperature. The final product powders are obtained by grinding the sintered pellets in a mortar. All synthesis operations are carried out under the protection of argon to prevent the materials from being attacked by moisture and oxygen in the ambient air.

Other experiments can be found in the Supplementary Information.

## Cell assembly

**steel | steel symmetric cells.** Symmetric steel|steel cells are employed to evaluate the ionic and electronic conductivity of the electrolytes. To fabricate steel|sulfide electrolytes|steel symmetric cells, sulfide powders are cold pressed into pellet at 300 MPa inside an 8 mm diameter PEEK die with two stainless-steel rods as current collectors.

**Li | Li symmetric cells.** Symmetric Li|Li cells are employed to evaluate the critical current densities, Li stripping/plating behavior of the electrolytes, and to observe the dendrite growth behavior/interfacial evolution through X-ray computed tomography (CT) or an operando optical microscopy observation device. To fabricate Li|Li symmetric cells for evaluating the critical current densities and Li stripping/plating behaviors, 80 mg sulfide powders are pressed

under 300 MPa inside an 8 mm diameter PEEK die to form a solid pellet. Then, two pieces of Li metal with thickness 50 µm and diameter 6 mm are placed onto each side of the sulfide pellet and following pressed by 120 MPa for 5 min to form the symmetrical cell. To assemble the tomography cell, one piece of lithium disc with thickness 100 µm and diameter 2.5 mm is firstly placed on a 3 mm diameter steel pin, which is interposed in a polyimide tube with diameter of 3 mm. Then, sulfide powders, another 2.5 mm diameter lithium disc on a steel pin are placed on top of each other. Afterwards, a pressure of ~110 MPa is applied to them. After releasing the pressure, the obtained symmetrical cell is sandwiched by two steel M5 bolts for testing. For the assembly of operando optical microscopy cell, sulfide powders are pressed under 300 MPa inside a half-round PEEK die to form a half-round solid pellet. Then, two pieces of half-round Li discs are placed onto both sides of the half-round sulfide pellet. Afterwards, a pressure of 120 MPa is applied to the three layers to form the half-round symmetrical cell.

**Al-GCs/VGCF | $Li_6PS_5Cl$ | Li/In batteries.** These batteries are assembled for CV testing to evaluate the electrochemical stable windows of Al-GCs. To this end, the cathode composites are firstly prepared by ball-milling Al-GCs and VGCF powders according to a weight ratio of 7:3 at 100 rpm for 2 h using an apparatus (FRITSCH, Pulverisette7). The electrolyte layer is prepared by pressing 80 mg $Li_6PS_5Cl$ powders at 200 MPa inside a 10 mm diameter PEEK die. Then, 10 mg of the cathode composites are dispersed on the surface of the as-prepared electrolyte pellet and pressed at 350 MPa. Finally, one piece of Li-In alloy foil is attached on the other side of the electrolyte pellet and pressed at 100 MPa.

**$LiCoO_2$ | Li or $LiCoO_2$ | Li/In and NCM811 | Li/In batteries.** The $LiCoO_2$ | Li or $LiCoO_2$ | Li/In and NCM811/Li/In batteries are prepared and tested as described below. The cathode composites are firstly prepared by manual grinding $LiNbO_3$ coated $LiCoO_2$ or bare-/$LiNbO_3$ coated-NCM811 and LPS7228 powders according to a weight ratio of 7:3. The electrolyte layer is prepared by pressing 80 mg electrolyte powders at 200 MPa inside a PEEK die (diameter of 10 mm). Then, 10 mg of the cathode composites are dispersed on the surface of the as-prepared electrolyte pellet and pressed at 350 MPa. Finally, one piece of lithium or indium foil (0.1 mm thickness, 8 mm diameter) is attached on the other side of the electrolyte pellet and pressed at 100 MPa. The $LiCoO_2$ or NCM811 loading is 8.92 mg cm$^{-2}$.

**Thick-cathode battery.** For thick-cathode cell, active material $LiNbO_3$ coated $LiNi_{0.6}Mn_{0.2}Co_{0.2}O_2$ and LPS7228 are uniformly mixed in a mass ratio of 80:20 to prepare the cathode composite. The fabricated LPS7228 film is cut into circular discs with 10 mm diameter. The mass of the used electrolyte film is around 8 mg. Afterwards, the LPS7228 film is placed into a 10 mm diameter PEEK die. Next, ~50 mg cathode composites are placed on one side of the film and µSi anode with diameter of 10 mm and active material of 2.2 mg is attached on the other side. The obtained sandwiched pellet is then pressed under 350 MPa with stainless steel as collectors on both sides.

## Characterization

**Cells or batteries.** An electrochemical working station (Biologic VMP-300) is used to perform the EIS experiments in a frequency range of 7 M to 100 mHz with an applied amplitude of 50 mV. DC polarization measurement is carried out to determine the electronic conductivity. Subsequently, time dependence of the current is measured under constant voltage 0.1 V for 2 h by using the electrochemical working station Biologic VMP-300. For CV testing, to probe the upper potential of the electrochemical stable window, the cell is cycled at a scan rate of 0.1 mV/s, starting at the open-circuit potential, up to 3.5 V and back to 1 V vs. Li$^+$/Li and the cycle ends at the open-circuit potential. To probe

the lower potential of the electrochemical stable window, the cell is cycled at a scan rate of −0.1 mV/s, starting at the open-circuit potential, down to 0.5 V and back to 2.0 V vs. Li$^+$/Li. Operando optical microscopy observation is performed in optical microscope (OLS-4000 OLYMPUS) at home-made device with a transparent window, where Li|Li symmetric cells with different electrolytes are assembled and the process of Li deposition at 0.5 mA cm$^{-2}$ is captured in real time. For the tomography cells, these cells after cycling at 0.5 mA cm$^{-2}$ are measured by laboratory X-ray CT instrument (Rigaku CT lab HX) under an automatic mode. The scan mode is High Resolution and the scan time is 68 min. During the measurement, the X-ray tube voltage is 70 kV and the tube current is 116 µA. The geometry between the sample and the detector is Long and the used field of view (FOV) is 10. The distance between Focus and Object is 32 mm and the distance between Focus and Detector is 432 mm. The resultant spatial resolution is 3.6 µm. The batteries are galvanostatically cycled using a Land battery test system (Land CT2001A, Wuhan Land Electronic Co. Ltd., China).

**Diffraction.** Lab X-ray diffraction (XRD) measurements are carried out using a Rigaku SmartLab X-ray diffractometer with Cu Kα radiation. The diffraction data are collected with a step width of 0.01° over a 2θ range of 10−80°. Synchrotron XRD measurements are performed using the BL01C2 beamline at Taiwan Synchrotron Radiation Research Center, with an X-ray wavelength of 0.82657 Å. The specimens are sealed under an Ar atmosphere in Lindemann glass capillaries (~0.3 mm inner diameter). Diffraction data are collected in 0.01° steps between 5 and 35° at 298 K. Rietveld refinements for the structural parameters are conducted using the GSAS II program[49]. Initially, the structural information obtained from the Materials Project database. These steps are used during the refinements: (i) scale factor, (ii) 12 coefficients for a Chebyshev function background, (iii) peak shape parameters, (iv) lattice parameters, (v) zero error, (vi) fractional atomic coordinates, (vii) atomic occupancies. Then, multiple correlated parameters are refined simultaneously, and the related constraints Tables can be found separately in Supporting Information.

**Raman spectra.** These are acquired using a micro-Raman spectrometer (Renishaw, inVia) equipped with a 532 nm Ar ion laser.

**Solid-state NMR.** Solid-state NMR experiments are performed on 19.96 T Varian VNMRS spectrometer ($^{31}$P Larmor frequency of 344.09 MHz) using 1.6 mm T3 MAS probe. $^{31}$P MAS NMR spectra are recorded using 90° pulse of 3 µs length and a recycle delay of 50 s. Prior to NMR experiments, the sample is packed into 1.6 mm Al-reduced zirconia rotor in Ar-filled glovebox. $N_2$ gas flow is used for MAS drive and bearing gas. The connectivity between phosphorus units is investigated by 2D $^{31}$P refocused INADEQUATE experiments. $^{31}$P chemical shifts are referenced to 85 % $H_3PO_4$ solution. All of the NMR data are processed with ssNake freeware[50].

**Cryogenic transmission electron microscope (cryo-TEM).** A TEM cryo-holder (Gatan 626) is used to load the sample where TEM grids are immersed in liquid nitrogen and then mounted onto the holder via a cryo-transfer workstation. The TEM images, high-resolution TEM (HRTEM) images and selected-area electron diffraction (SAED) patterns are obtained using Tecnai G$^2$ F20 (FEI) at an acceleration voltage of 200 kV.

**off-axis Electron holography.** Electron hologram image is obtained by using FEI-Themis Z with High brightness Schottky field emission electron source and an accelerating voltage of 80 kV.

**X-ray photoelectron spectroscopy (XPS).** The samples are placed onto double-sided tape fixed to clean glass slides and placed in a vacuum transfer holder inside an Ar-filled glovebox. An X-ray

photoelectron spectroscopy spectrometer (Thermo Scientific Model K-Alpha XPS) with a monochromatized Al Kα source (1486.6 eV) is used to obtain the surface chemistry of the samples. A 400 μm X-ray spot size is used to maximize the signal intensity and to obtain an average surface composition over a large area. The sample surface is cleaned via Ar⁺ sputtering with an acceleration voltage of 0.5 kV for 200 s. A dual beam charge neutralization is applied for charge compensation. The base pressure in the analysis chamber is $3 \times 10^{-10}$ mbar. The pass energy is 23.5 eV. Spectra are charge corrected using the C 1 s core level peak set to 284.8 eV. Data evaluation is performed with the software Thermo Avantage XPS software package (version 5.976).

**Computation method.** We employ the Vienna Ab Initio Package (MedeA-VASP5.4)[51,52] to perform all the DFT calculations within the GGA using the PBE formulation[53]. We chose the projected augmented wave (PAW) potentials[54,55] to describe the ionic cores and take valence electrons into account using a plane wave basis set with a kinetic energy cutoff of 520 eV. Partial occupancies of the Kohn−Sham orbitals are allowed using the Gaussian smearing method and a width of 0.05 eV. The electronic energy is considered self-consistent when the energy change is smaller than $10^{-5}$ eV. A geometry optimization is considered convergent when the force change is smaller than 0.01 eV/Å. At low temperature, the entropic contributions to Gibbs free energy change ($\Delta G_r$) are small, and the volume change of solid material is relatively small which could be ignored, so the $\Delta G_r$ can be approximated by the internal energy change ($\Delta E_r$), $\Delta G_r \approx \Delta E_r$, which can be obtained from DFT calculations[56].

According to previous reports, the most common bare surfaces are (100) and (010) for $Li_{2.82}Al_{0.06}PS_4$ and $Li_7P_3S_{11}$, respectively[57,58]. To obtain the most stable interface structures, their structure optimizations are performed by fixing the bottom 2 layers and 1 layer for $Li_7P_3S_{11}(010)@Li_{2.82}Al_{0.06}PS_4$ (100) and $Li_7P_3S_{11}(010)@Li_7P_3S_{11}(010)$, respectively, meanwhile allowing other atomic positions to vary. The k-point meshes of $2 \times 3 \times 4$ and $4 \times 2 \times 2$ are used for Brillouin zone (BZ) sampling for $Li_{2.82}Al_{0.06}PS_4$ and $Li_7P_3S_{11}$, respectively. Four interface structures are constructed in Supplementary Fig. 6a. The binding energy ($E_{bind}$) is evaluated the $Li_7P_3S_{11}$ nucleation on the different surfaces, which is defined as the energy that different layers of $Li_7P_3S_{11}(010)$ are formed on the $Li_{2.82}Al_{0.06}PS_4(100)$ or $Li_7P_3S_{11}(010)$ surface. The $E_{bind}$ is formulated as:

$$E_{bind} = \frac{1}{A}\left(E_{interface} - E_{Li_7P_3S_{11}} - E_{Li_{2.82}Al_{0.06}PS_4}\right)$$

where $E_{interface}$ represents the total energy of interface structure. $E_{Li_7P_3S_{11}}$ and $E_{Li_{2.82}Al_{0.06}PS_4}$ represent the energy of $Li_7P_3S_{11}(010)$ and $Li_{2.82}Al_{0.06}PS_4(100)$, respectively. $A$ is the surface area which we are interesting.

The grain boundaries structures which contain single-crystal structures on either side are created using the coincident site lattice (CSL) theory[59]. During structural optimizations, the $2 \times 2 \times 1$ Monkhorst-Pack k-point grid for Brillouin zone is used for $Li_7P_3S_{11}/Li_7P_3S_{11}$ (010) grain boundary structures. Finally, nudged elastic band (NEB) method has been employed to calculate the Li ions migration along the grain boundary and across the grain regions.

The Li chemical potential is the minus of the Li-vacancy formation energy with respect to Li metal ($\mu_{Li} = -E_v$), which is defined as:

$$E_v(Li_i) = [E(Li_i) + \mu_{Li(metal)}] - E$$

where the $E(Li_i)$ and $E$ are the energies of the $Li_{2.82}Al_{0.06}PS_4$ or $Li_7P_3S_{11}$ with and without defect $Li_i$, respectively; and $\mu_{Li(metal)}$ is the Li chemical potential in Li metal with a body-centered cubic-type structure.

## Data availability
The relevant data generated in this study are provided in the Supplementary Information/Source Data file. Source data are provided with this paper.

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

## Acknowledgements

This work was supported the Strategic Priority Research Program of the Chinese Academy of Sciences (XDA22010600), the National Nature Science Foundation of China (51902325, 41907105), the Key Scientific and Technological Innovation Project of Shandong (2020CXGC010401), and Natural Science Foundation of Shandong Province (ZR2021ZD25) and QIBEBT(SZ202101). H.Q. acknowledged the financial support for China Scholarship Council (Grant No. 201804910640). The Netherlands Organisation for Scientific Research (NWO) for advanced materials

science is greatly acknowledged for the support of the solid-state NMR facility which is part of the uNMR-NL ROADMAP facilities (NWO project no. 184.035.002). We acknowledged support from the Max Planck-POSTECH-Hsinchu Center for Complex Phase Materials.

## Author contributions

Y.W., H.Q., B.L., and X.L. contributed equally to the paper. G.C. and J.J. proposed the concepts. Y.W. designed and carried out the experiments. B.L., X.L., and C.L. conducted the cryo-TEM studies. H.Q., E.v., and A.K. conducted the SSNMR studies. Z.H., C.C., and H.S. conducted the SXRD studies. S.Z. performed the theoretical simulations. Y.W., J.L., L.Cui, and F.J. analyzed the data. J.J. supervised the research. Y.W. wrote the manuscript with the help from J.J., J.M. and L.Chen. All authors discussed the results and commented on the manuscript.

## Competing interests

The authors declare no competing interests.
