## [Peer Review File · Nature Communications]

REVIEWER COMMENTS

Reviewer #1 (Remarks to the Author):

The manuscript “Self-organized hetero-nanodomains actuating super Li⁺ conduction in glass ceramics” has presented super Li⁺-conductive glass ceramics which can self-organize into hetero-nanodomains during the solid-state reaction. The performance is impressive, and the authors have presented detailed discussion about the mechanism from experimental and DFT results.

This work is very interesting and important, I recommend accepting this manuscript after the authors explaining some questions listed below:

1. In page 8, line 181, “By excluding the unreacted Al₂S₃”, how to achieve that? Is this step necessary before other electrochemical tests of the electrolyte?
2. The conductivity changes sharply with small differences in Li₂S:P₂S₅ ratio (especially from 72 to 71), could the authors provide a clearer explanation?
3. Activation energy could be calculated from the conductivity under various temperature, please show the experimental results and compared with the DFT results.
4. The electrochemical stability towards Li metal is very good, what about the moisture stability? A comparison of H₂S amount by exposing LPS70-75 in moist air would be helpful.
5. Please show the comparison of electrochemical windows of the mentioned electrolytes. Is it possible to match the electrolyte with high-voltage NCM811?
6. The flexibility and high conductivity of LPS7228 film is interesting. Would the conductivity decrease after bending? Please show the results.
7. What’s the high-temperature performance of the mentioned sulfide electrolyte and batteries? In many works, 60°C is always applied for better performance. Would the long-time heating harmful to the hetero-nanodomains (e.g. making them homogeneous)?

Reviewer #2 (Remarks to the Author):

The authors proposed a nanocrystallization strategy to fabricate super Li⁺-conductive glass ceramics via regulating the nucleation energy; the crystallites within glass ceramics can self-organize into

hetero-nanodomains during the solid-state reaction, and thus, glass ceramics achieved Li⁺ conductivity of 13.2 mS cm⁻¹. The high Li⁺ conductivity ensures stable operation of a 220 μm thick LiNi_{0.6}Mn_{0.2}Co_{0.2}O₂ composite 33cathode (8 mAh cm⁻²), with which the all-solid-state lithium battery reaches a high energy density of 34420 Wh kg⁻¹ by cell mass and 834 Wh L⁻¹ by cell volume at room temperature. Although the route to getting high Li⁺ conductivity is new, and the results are promising, many loopholes still need to be filled and addressed. After addressing the question, I would like to accept this article for nature communication.

My comments are given below.

Comment#1. According to Figure 1c, the authors claimed that the large proportion of grain boundaries (12 grain boundaries (red) resulting from two adjacent 107 Crys2 nanocrystallites, 8 grain boundaries (green) resulting from three adjacent Crys2 108 nanocrystallites, and 6 grain boundaries (violet) between Crys1 and Crys2 nanocrystallites) occurring in the self-organized hetero-nanodomains are predicted to significantly enhance the σ_{Li^+} of GC. However, the advanced studies recognized that a large proportion of grain boundaries increase the grain boundary and then bulk resistance, which has an adverse impact on σ_{Li^+} of GC. Moreover, how did the authors exclude grain boundary resistance's influence in the self-organized hetero-nanodomains? Need to be explained with solid references.

Comment#2. The size of self-organized hetero-nanodomains needs to be measured.

Comment#3. PDF of standard Li₃PS₄ needs to be provided in Figure S1a for SXRD patterns. With increasing the nucleation-accelerant "Al₂S₃" phases, e.g., P2S74- and P2S64- are produced in GCs. Why? The kind of bonding (physical/chemical) that the "Al" has in the crystal structure of Li₃PS₄ is not clear from Raman and 31P MAS NMR spectra. So, XPS/27Al SSNMR needs to be conducted for deep crystal structure analysis.

Note! Reference 23 should be removed as it is not published yet. Qu, H. et al. Aluminum Ion Doping Mechanism of Lithium Thiophosphate Solid Electrolytes Revealed with Solid-state NMR. Submitted (2022)

Comment#4. Figure 2a 31P MAS NMR spectra show that impure phase P2S74- and P2S64- are produced with an increase of nucleation-accelerant "Al₂S₃" and identified for optimized LP7228. However, the emergence of P2S64- in Li₃PS₄ GC can reduce the σ_{Li^+} to a large extent. For example, how can the authors exclude the impact of impure phases to get a high σ_{Li^+} of 13.2 mS cm⁻¹? Moreover, the Arrhenius plots and the activation energies should be given for related compositions because a more significant ionic conductivity does not always mean smaller activation energy.

Comment#5. Cyclic voltammetry is an essential measurement to determine the electrochemical voltage window of solid electrolytes. The authors should test cyclic voltammetry by the new method in Adv. Energy Mater. 2016, 6, 1501590 and then compare the results of LPS7228 to the pristine electrolyte.

Comment#6. The sulfide SSEs are reported to be reduced at the Li-metal anode side. Does the incorporation of Al₂S₃ prevents the growth of lithium dendrites or suppress the interfacial reaction?

Comment#7. The value of critical current density (CCD) has been widely adopted to quantify the dendrite suppression capability of the SSEs. Thus, the CCD of all the composition need to be reported as given in the recently published reference; (Adv. Funct. Mater. 2022, 32, 2201528). Also, provide the galvanostatic cycling performance of Li//Li symmetric cells with counterparts.

Comment#8. From Figure S13, bulk resistance of Li|LPS7228|Li symmetric cell after 500 cycles is relatively lower (similar to 1st cycle) than after 200 and 1000 cycles, respectively. What happens inside the cell that reduces the bulk resistance of the Li|LPS7228|Li symmetric cell? Explain the reason.

Comment#9. Low electronic conductivity is essential to suppress the Li-dendrite formation inside the SSE. Thus, the electronic conductivity of solid electrolytes needs to be measured.

Comment#10. It is essential to provide more basic electrochemical properties of LCO/LPS7228/Li battery, e.g., cyclic voltammetry, long cycling performance under high current density, e.g., 0.5 C and EIS before and after measurements.

Reviewer #3 (Remarks to the Author):

Today it is widely acknowledged that the low ionic conductivity of solid electrolytes along with its interfacial instability are the main obstacles in safer solid state battery development. The manuscript shows the way of increasing Li₂S-P₂S₅ glass-ceramics ionic conductivity by tuning the microstructure. The authors use the addition of various sulphides as a tool for controlling the composition and microstructure of the obtained solid electrolytes. Unlike many other papers in the field, here author show the clear idea rather than going through simple trial-and-error route. I think the report is suitable for publishing in Nature communications, however, some point should be addressed by the authors before acceptance of the manuscript:

- The authors claim that the interface between lithium and glass-ceramics with Al₂S₃ added and the optimal composition stays smooth during Li plating/stripping. First, it is not clear what happens with the interface when other sulfide modifiers are used. Second, I'm wondering whether the interface with argyrodite can be stable if Li-Al alloy will be used instead of pure Li. Is there any difference between in situ formed Li-Al layer and Li-Al alloy used as anode? To the best of my knowledge, Li-Al alloy foils undergo severe dendrite formation during cycling as well. Third, I think the XPS data in Fig. S16 and its discussion are not convincing. I found no experimental details on XPS measurements which are very important as the electrolyte sample are electronic insulators thus interpretation of the spectra can be strongly affected by charging and differential charring effects.

- The understanding of the phase composition and evolution in the system is of a high importance. The authors put a lot of efforts in the XRD analysis. However, there are no details on Reitveld

refinements given in the Supplementary info - no structural models, no refinement algorithm (sequence for refining different parameters), etc.

- The authors say that "a 2D 31P refocused INADEQUATE experiment was performed for LPS7228". I can not understand this sentence. Please rephrase.

- When the authors describe the compositions of the glass-ceramics, the sum of mass concentrations is higher than 100%. While reading it becomes more clear what do the authors mean. However, I suggest more straightforward description of the compositions through all the manuscript and supplementary info.

Responses to the reviewers' comments

Dear reviewers,

On behalf of my co-authors, we appreciate the reviewers very much for your constructive comments on our manuscript entitled “*Self-organized hetero-nanodomains actuating super Li⁺ conduction in glass ceramics*” (No. NCOMMS-22-28440-T). These comments are all valuable and very helpful for revising and improving our manuscript. The manuscript has been revised carefully according to the comments and suggestions. All the modifications are highlighted. Below is a detailed description of the changes and the point-to-point response to the reviewers' comments.

With best regards,

Yours sincerely,

Arno P. M. Kentgens & Guanglei Cui

List of changes made

The followings are the main changes we made in the revision.

1. Author Jiedong Li is added in the revision. He tests the temperature-dependent electrochemical impedance spectroscopy of Al-GCs.
2. A new fund ‘Natural Science Foundation of Shandong Province (ZR2021ZD25)’ is added to section ACKNOWLEDGEMENTS.
3. In the revised manuscript, two sections of DATA AVAILABILITY and AUTHOR CONTRIBUTIONS are added.
4. In order to make the experimental methods more clear, we supplement the relevant experimental details in the METHOD section. All changes have been highlighted.
5. In the revised manuscript, Reference 23 in the original revision is removed, and the references’ orders are changed accordingly.
6. In the revised Supporting Information, Table S2-1–S2-9 are newly added.
7. In the revised Supporting Information, Table S2, S3, S4 in the original version is changed to Table S3, S4, S5, respectively.
8. In the revised Supporting Information, Figure S1a is modified.
9. In the revised Supporting Information, Figure S8 & S9 with related analysis are newly added.
10. In the revised Supporting Information, Figure S8, S9, S10, S11 in the original version are changed to Figure S10, S11, S12, S13, respectively.
11. In the revised Supporting Information, Figure S14 with related analysis is newly added.
12. In the revised Supporting Information, Figure S12, S13, S14, S15, S16 in the original version are changed to Figure S15, S16, S17, S18, S19, respectively.
13. In the revised Supporting Information, Figure S20 with related analysis is newly added.
14. In the revised Supporting Information, Figure S17 in the original version is changed to Figure S21.
15. In the revised Supporting Information, Figure S22 with related analysis is newly added.
16. In the revised Supporting Information, Figure S18 & S19 in the original version are changed to Figure S23, S24, respectively.

Reviewer #1:

General Comments: *The manuscript “Self-organized hetero-nanodomains actuating super Li⁺ conduction in glass ceramics” has presented super Li⁺-conductive glass ceramics which can self-organize into hetero-nanodomains during the solid-state reaction. The performance is impressive, and the authors have presented detailed discussion about the mechanism from experimental and DFT results. This work is very interesting and important, I recommend accepting this manuscript after the authors explaining some questions listed below.*

Response: We feel great thanks for your time and professional review work on our article. According to your encouraging suggestions, we have made modifications/corrections to our previous draft. The revisions are highlighted in the manuscript clearly.

Comment 1 *In page 8, line 181, “By excluding the unreacted Al₂S₃”, how to achieve that? Is this step necessary before other electrochemical tests of the electrolyte?*

Response: Due to our unclear expression, you misunderstood this sentence. Actually, ‘exclusion of the unreacted Al₂S₃’ does NOT mean to experimentally eliminate the unreacted Al₂S₃ reagent from the products, but to do so theoretically for the purpose of estimating the stoichiometry of LPS7228. In order not to cause any confusion to the readers, we have rephrased the wording of line 182 to: “By theoretically excluding the unreacted Al₂S₃”.

Changes in the revised manuscript: Page 8, line 182, “By theoretically excluding the unreacted Al₂S₃”

Comment 2 *The conductivity changes sharply with small differences in Li₂S:P₂S₅ ratio (especially from 72 to 71), could the authors provide a clearer explanation?*

Response: There are two main reasons for the conductivity changes from LPS7228 to LPS7129. According to the species identification (**Figure S1**), LPS7129 involves 27.4 wt % Li₄P₂S₆ phase, which is an extremely poor Li⁺ conductor with conductivity of ~ 10⁻⁷ S cm⁻¹ at room temperature (*J. Power Sources*, 2006, 159, 193–199; *Solid State Ionics*, 2016, 284, 61–70), while no Li₄P₂S₆ phase is detected in LPS7228. On the other hand, as shown in **Figure R1** cryo-TEM of LPS7129 does not show adequate nanodomains which have been verified by this work to be highways for fast Li⁺ migration. As a consequence, LPS7129 exhibits one order of magnitude lower conductivity than that of LPS7228.

Figure R1. Cryo-TEM of LPS7129.

Comment 3 *Activation energy could be calculated from the conductivity under various temperature, please show the experimental results and compared with the DFT results.*

Response: According to your great suggestion, temperature-dependent electrochemical impedance spectroscopy (EIS) measurements are conducted to derive the Arrhenius plots of Al-GCs (**Figure R2a**). The activation energies (E_a) of Al-GCs are determined based on the Arrhenius equation:

$$\sigma = \frac{\sigma_0}{T} \exp\left(\frac{-E_a}{k_B T}\right) \quad \text{Equation (R1)}$$

where, σ , σ_0 , T , k_B denote the Li^+ conductivity, exponential prefactor, absolute temperature, Boltzmann constant, respectively.

As shown in **Figure R2b**, a value of 0.310 eV is reached for LPS7525 and decreases to 0.305 eV for LPS7426. With decreasing $\text{Li}_2\text{S} : \text{P}_2\text{S}_5$ ratio, E_a decreases to 0.304 eV for LPS7327, and reaches a minimum of 0.291 eV for LPS7228. The decrease of E_a may be contributed to the formation of $\text{Li}_{3-3x}\text{Al}_x\text{PS}_4$ with increased carrier's density and hetero-nanodomains. Further reducing the $\text{Li}_2\text{S} : \text{P}_2\text{S}_5$ ratio increases E_a to 0.316 eV (LPS7129), 0.352 eV (LPS7030), 0.375 eV (LPS6931) and 0.398 eV (LPS6535). According to the species evolution of Al-GCs, further decreasing Li_2S content generates large amount of poor Li^+ conductor of $\text{Li}_4\text{P}_2\text{S}_6$ ($E_a = 0.46$ eV, *Chem. Mater.* 2016, 28, 23, 8764–8773) and even non- Li^+ conductor aluminum thiophosphates, blocking the Li^+ migration and thus leading to a large E_a .

For DFT results, E_a is 0.21 eV when Li^+ migrating along grain boundary and 0.37 eV when Li^+ migrating across bulk (grain). Given that the experimental E_a is derived from both the Li^+ migration along grain boundary and across bulk, the value 0.291 eV (LPS7228) agrees well with the DFT results.

Figure R2. (a) Arrhenius plots of the conductivity for Al-GCs. (b) Activation energy for Li^+ conduction of Al-GCs, as calculated from Arrhenius plots.

Changes in the Supporting Information related to Figure S8 in the revised version: “Temperature-dependent electrochemical impedance spectroscopy (EIS) are conducted to derive the Arrhenius plots of Al-GCs (Figure S8a below). The activation energies (E_a) of Al-GCs are determined based on the Arrhenius equation:

$$\sigma = \frac{\sigma_0}{T} \exp\left(\frac{-E_a}{k_B T}\right) \quad \text{Equation (S1)}$$

where, σ , σ_0 , T , k_B denote the Li^+ conductivity, exponential prefactor, absolute temperature, Boltzmann constant, respectively.

As shown in Figure S8b, a value of 0.310 eV is reached for LPS7525 and decreases to 0.305 eV for LPS7426. With decreasing $\text{Li}_2\text{S} : \text{P}_2\text{S}_5$ ratio, E_a decreases to 0.304 eV for LPS7327, and reaches a minimum of 0.291 eV for LPS7228. The decrease of E_a may be contributed to the formation of $\text{Li}_{3-3x}\text{Al}_x\text{PS}_4$ with increased carrier’s density and hetero-nanodomains. Further reducing the $\text{Li}_2\text{S} : \text{P}_2\text{S}_5$ ratio increases E_a to 0.316 eV (LPS7129), 0.352 eV (LPS7030), 0.375 eV (LPS6931) and 0.398 eV (LPS6535). According to the species evolution of Al-GCs, further decreasing Li_2S content generates large amount of poor Li^+ conductor of $\text{Li}_4\text{P}_2\text{S}_6$ ($E_a = 0.46$ eV) and even non- Li^+ conductor aluminum thiophosphates, blocking the Li^+ migration and thus leading to a large E_a ”

Comment 4 The electrochemical stability towards Li metal is very good, what about the moisture stability? A comparison of H_2S amount by exposing LPS70-75 in moist air would be helpful.

Response: Thank you for the suggestion. To evaluate the moisture stability of Al-GCs, total generation amount of H_2S is characterized by on-line gas analysis mass spectrometry (MS) system (HPR-20, Hiden Analytical Ltd.). A saturated KI solution (25 °C) is used to simulate a room environment with a 68 % RH (relative humidity). As shown in Figure R3a, 150 mg Al-GCs powders are placed in a glass

bottle with continuous flow of humid argon gas. The generated H₂S gas is instantly blown out by the flowing argon gas and then detected by the MS system (**Figure R3b**). In addition, dry argon gas is used to clean the entire pipeline before the experiment. **Figure R3c** shows the curves of the ionic current of H₂S as a function of exposure duration, while the integral area of the curves is proportional to the total generation amount of H₂S. As shown in **Figure R3c**, H₂S is generated once the Al-GCs are exposed to moist argon gas, and the total generation amount of H₂S continuously increased with decreasing the Li₂S content.

Figure R3. (a) Schematic illustration of on-line H₂S analysis MS system. (b) Image of the on-line H₂S analysis MS system. (c) The total H₂S generation amount of Al-GCs as a function of exposure time.

Comment 5 Please show the comparison of electrochemical windows of the mentioned electrolytes. Is it possible to match the electrolyte with high-voltage NCM811?

Response: Experimentally, the electrochemical window (EW) is usually probed via the cyclic voltammetry (CV) method. According to the approach reported by Han *et al* (*Adv. Energy Mater.*, 2016, 6, 1501590), an asymmetric cell setup ((Al-GCs)-VGCF|Li₆PS₅Cl|Li/In) with lithium-indium alloy (acting as both counter and reference electrode) and a composite of Al-GCs with VGCF (weight ratio of Al-GC to VGCF is 70 : 30) is used as working electrode to obtain the intrinsic EW of Al-GCs.

Specifically, to probe the upper potential of the EW, the cell is cycled at a scan rate of 0.1 mV/s, starting at the open-circuit potential, up to 3.5 V and back to 1 V vs. Li⁺/Li and the cycle ends at the open-circuit potential. To probe the lower potential of the EW, the cell is cycled at a scan rate of -0.1

mV/s, starting at the open-circuit potential, down to 0.5 V and back to 2.0 V vs. Li⁺/Li. The EWs of Al-GCs extracted from **Figure R4a** and **R4b** are shown in **Figure R4c**. The EWs of Al-GCs are 2.02–2.7, 2.03–2.77, 1.94–2.93, 1.53–2.94, 1.59–2.99, 1.57–2.97, 1.47–2.95, 1.48–3.01 V vs. Li⁺/Li for LPS7525, LPS7426, LPS7327, LPS7228, LPS7129, LPS7030, LPS6931, LPS6535, respectively.

To test whether Al-GCs can match NCM811, all-solid-state lithium batteries (ASSBs) using LPS7228 (30 wt%) + bare- or LiNbO₃ coated NCM811 (70 wt%) cathode composite, Li₆PS₅Cl electrolyte and Li/In anode were fabricated and cycled at 0.1 C rate between 3.0 and 4.5 V vs. Li⁺/Li. As shown in **Figure R4d** and **R4e**, the bare-NCM811|Li₆PS₅Cl|Li/In battery exhibits fast capacity fade with 52.5 % capacity retention over only 70 cycles, similar to the results reported by Koerver *et al* (*Chem. Mater.*, 2017, 29, 5574–5582). This common phenomenon is caused by the narrow thermodynamic intrinsic EW of sulfide electrolytes ranging from 1.7 to 2.9 V vs. Li⁺/Li (*Chem. Mater.*, 2019, 31, 8328–8337; *Nat. Mater.*, 2020, 19, 428–435), and the tendency for the high-voltage cathode oxidizing the sulfide electrolytes in physical contact, in particular at high charging potential (*Nano Lett.*, 2020, 20, 1483–1490; *Nat. Energy*, 2022, 7, 83–93).

Coating cathode active material particles with an electronically insulating/ionically conductive, chemically compatible material has been verified to address this problem effectively (*ACS appl. Mater. Interfaces*, 2021, 13, 41669–41679; *Adv. Energy Mater.*, 2021, 2100126). Accordingly, the ASSB with LiNbO₃ coated NCM811 cathode composite (**Figure R4f**) demonstrates much better electrochemical performance, maintaining 80 % capacity retention over 70 cycles. **Consequently, it can be concluded that although LPS7228 has a narrow EW, it can still match high-voltage cathode materials with stable coating.**

Figure R4. CV of (Al-GCs)-VGCF|Li₆PS₅Cl|Li/In cells at a scan rate of 0.1 mV/s in the voltage range of (a) 1.0 – 3.5 V and (b) 0.5 – 2.0 V. (c) Practical EW of Al-GCs estimated from (Al-GCs)-VGCF|Li₆PS₅Cl|Li/In cells. The stability region for each electrolyte is shown in orange. (d) Cycle behavior of bare-NCM811|Li₆PS₅Cl|Li/In and LiNbO₃ coated NCM811|Li₆PS₅Cl|Li/In ASSB at 0.1 C and room temperature. (e) Charge-discharge curves of the LiNbO₃ coated NCM811|Li₆PS₅Cl|Li/In ASSB at 0.1 C and room temperature. (f) HRTEM image of LiNbO₃ coated NCM811 particle and Ni, Co, Mn, Nb elemental maps.

Changes in the Supporting Information related to Figure S20 in the revised version:

“Experimentally, the electrochemical window (EW) is usually probed via the cyclic voltammetry (CV) method. According to the approach reported by Han *et al.*, an asymmetric cell setup ((Al-GCs)-VGCF|Li₆PS₅Cl|Li/In) with lithium-indium alloy (acting as both counter and reference electrode) and a composite of Al-GCs with VGCF (weight ratio of Al-GC to VGCF is 70 : 30) is used as working electrode to obtain the intrinsic EW of Al-GCs.

Specifically, to probe the upper potential of the EW, the cell is cycled at a scan rate of 0.1 mV/s, starting at the open-circuit potential, up to 3.5 V and back to 1 V vs. Li⁺/Li and the cycle ends at the open-circuit potential. To probe the lower potential of the EW, the cell is cycled at a scan rate of –0.1 mV/s, starting at the open-circuit potential, down to 0.5 V and back to 2.0 V vs. Li⁺/Li. The EWs of Al-GCs extracted from Figure S20a and S20b are shown in Figure S20c. The EWs of Al-GCs are 2.02–2.7, 2.03–2.77, 1.94–2.93, 1.53–2.94, 1.59–2.99, 1.57–2.97, 1.47–2.95, 1.48–3.01 V vs. Li⁺/Li for LPS7525, LPS7426, LPS7327, LPS7228, LPS7129, LPS7030, LPS6931, LPS6535, respectively.

To test whether Al-GCs can match high-voltage NCM811, all-solid-state lithium batteries (ASSBs) using LPS7228 (30 wt%) + bare- or LiNbO₃ coated NCM811 (70 wt%) cathode composite, Li₆PS₅Cl electrolyte and Li/In anode were fabricated and cycled at 0.1 C rate between 3.0 and 4.5 V vs. Li⁺/Li. As shown in **Figure S20d** and **S20e**, the bare-NCM811|Li₆PS₅Cl|Li/In battery exhibits fast capacity fade with 52.5 % capacity retention over only 70 cycles, similar to the results reported by Koerver *et al.* This common phenomenon is caused by the narrow thermodynamic intrinsic EW of sulfide electrolytes ranging from 1.7 to 2.9 V vs. Li⁺/Li, and the tendency for the high-voltage cathode oxidizing the sulfide electrolytes in physical contact, in particular at high charging potential.

Coating cathode active material particles with an electronically insulating/ionically conductive, chemically compatible material has been verified to address this problem effectively. Accordingly, the ASSB with LiNbO₃ coated NCM811 cathode composite (**Figure S20f**) demonstrates much better electrochemical performance, maintaining 80 % capacity retention over 70 cycles. Consequently, it can be concluded that although LPS7228 has a narrow EW, it can still match high-voltage cathode materials with stable coating. It should be noted that the cathode active materials used in this work are all coated with LiNbO₃.

Comment 6 *The flexibility and high conductivity of LPS7228 film is interesting. Would the conductivity decrease after bending? Please show the results.*

Response: Here, to test whether the conductivity of LPS7228 film decreases after bending, we fix one side of an LPS7228 film on a glass bottle with a diameter of 2.76 cm, and hold the other side with tweezers for bending test in a glovebox filled with argon (**Figure R5b**). Then, the conductivity of the LPS7228 film after the 10th and 20th bending are evaluated. As shown in **Figure R5e**, the LPS7228 film shows an ionic conductivity of 8.69 and 8.40 mS cm⁻¹ after the 10th and 20th bending, respectively, which are slightly lower than that of the pristine LPS7228 film (10.1 mS cm⁻¹). The slight decrease in conductivity may be due to the formation of micro-cracks within the LPS7228 film caused by bending.

Figure R5. (a) Image of an LPS7228 film with a thickness of 110 μm and an area of $2.5 \times 5.25 \text{ cm}^2$. Images of the LPS7228 film (b) before bending, (c) after the 10th bending and (d) after 20th bending. (e) EIS plots at room temperature of the pristine LPS7228 film and after the 10th, 20th bending.

Comment 7 *What's the high-temperature performance of the mentioned sulfide electrolyte and batteries? In many works, 60°C is always applied for better performance. Would the long-time heating harmful to the hetero-nanodomains (e.g. making them homogeneous)?*

Response: That is a good question. As for the electrolytes, the σ_{Li^+} of LPS7228 powders and LPS7228 film reaches 35.5 and 25.8 mS cm^{-1} at 60 °C, respectively (**Figure R6d**).

D.C. galvanostatic test is conducted to further evaluate the compatibility of LPS7228 with lithium metal anode at 60 °C. **Figure R6a** displays an overpotential gradually increases from 4.8 to 5.6 mV during the first 20 hrs at 0.1 mA cm^{-2} /0.1 mA h cm^{-2} , which corresponds to the formation of a stabilized Li/LPS7228 interface. Subsequently, this stability interface can support a smooth Li plating/stripping process for over 300 hrs at 0.2 mA cm^{-2} / 0.2 mA h cm^{-2} (this battery is still cycling steadily). Compared with the symmetric cells at room temperature (**Figure 4a** in the manuscript), high temperature significantly improves the kinetics of ion transports and Li plating/stripping, leading to a lower overpotential.

Then, the LiNbO_3 -coated NCM622|LPS7228 film| μSi battery with an areal loading of 44.6 mg cm^{-2} operates in a voltage range of 2.6 – 4.4 V at 60 °C. As shown in **Figure R6b**, an initial discharge capacity of 175.8 mAh g^{-1} is obtained at 0.1 C, corresponding to an areal capacity of 7.84 mAh cm^{-2} . Due to the improved kinetic at 60 °C, a high value of 6.97 mAh cm^{-2} is still obtained when increasing the C-rate to 0.2 C. Remarkably, a high reversible areal capacity of 6.04 mAh cm^{-2} is maintained after 100 cycles, showing a good retention of 86.6 % with an average Coulombic efficiency (CE) of 99.5 %. Compared with the performance at room temperature (**Figure 4j**), higher discharge capacity and

capacity retention can be achieved at 60 °C.

Also, the high-temperature performance of LiNbO₃-coated LiCoO₂|LPS7228|Li is evaluated at 60 °C, as shown in **Figure R6c**. Due to the high σ_{Li^+} and anodic stability of LPS7228, a reversible specific discharge capacity of 143.1 mAh g⁻¹ is achieved for the first cycle with high initial of CE 96.8 %, and maintains 132.9 mAh g⁻¹ after 100 cycles at 0.2 C, showing a good retention of 92.8 %, which is better than that of cycling at room temperature.

To verify whether the long-time heating is harmful to the hetero-nanodomains, the LPS7228 electrolytes are extracted from the LiNbO₃-coated LiCoO₂|LPS7228|Li ASSB after 100 cycles at 60 °C (heat treated at 60 °C for 1061 hours). As for the σ_{Li^+} , the heated-treated LPS7228 still remains a high value of 34.8 mS cm⁻¹ at 60 °C, which is almost identical to that of the pristine LPS7228, 35.5 mS cm⁻¹ (**Figure R6d**). To check whether long-time heating is harmful to the hetero-nanodomains (*e.g.*, making them homogeneous), the microstructure of heated treated LPS7228 is observed by cryo-TEM. As shown in **Figure R6e** and **R6f**, a large amount of hetero-nanodomains still exist. Consequently, it can be concluded that the long-time heating can not damage the hetero-nanodomains, which are self-organized when annealing at 300 °C as indicated by *in situ* temperature-dependent XRD (**Figure S5**).

Figure R6. (a) Galvanostatic cycling measurements of Li|LPS7228|Li at 0.1 mA cm⁻²/0.1 mAh cm⁻², 0.2 mA cm⁻²/0.2 mAh cm⁻² and 60 °C. Cycle behavior of (b) NCM622|LPS7228 film|μSi and (c) LiCoO₂|LPS7228|Li ASSB at 60 °C. (d) EIS plots at 60 °C of LPS7228 film, LPS7228 extracted from the as-cycled battery at 60 °C after 100th

cycling for 1061 h and pristine LPS7228. (e) Cryo-TEM and (f) SAED of LPS7228 extracted from the as-cycled battery at 60 °C after 100th cycling.

Reviewer #2:

General Comments: *The authors proposed a nanocrystallization strategy to fabricate super Li⁺conductive glass ceramics via regulating the nucleation energy; the crystallites within glass ceramics can self-organize into hetero-nanodomains during the solid-state reaction, and thus, glass ceramics achieved Li⁺ conductivity of 13.2 mS cm⁻¹. The high Li⁺ conductivity ensures stable operation of a 220 μm thick LiNi_{0.6}Mn_{0.2}Co_{0.2}O₂ composite cathode (8 mAh cm⁻²), with which the all-solid-state lithium battery reaches a high energy density of 420 Wh kg⁻¹ by cell mass and 834 Wh L⁻¹ by cell volume at room temperature. Although the route to getting high Li⁺ conductivity is new, and the results are promising, many loopholes still need to be filled and addressed. After addressing the question, I would like to accept this article for nature communication.*

Response: We appreciate the reviewer for spending time to review, and we hope our responses satisfy the reviewer. Please find the point-by-point responses to the comments below.

Comment 1 *According to Figure 1c, the authors claimed that the large proportion of grain boundaries (12 grain boundaries (red) resulting from two adjacent Crys2 nanocrystallites, 8 grain boundaries (green) resulting from three adjacent Crys2 nanocrystallites, and 6 grain boundaries (violet) between Crys1 and Crys2 nanocrystallites) occurring in the self-organized hetero-nanodomains are predicted to significantly enhance the σ_{Li^+} of GC. However, the advanced studies recognized that a large proportion of grain boundaries increase the grain boundary and then bulk resistance, which has an adverse impact on σ_{Li^+} of GC. Moreover, how did the authors exclude grain boundary resistance's influence in the self-organized hetero-nanodomains? Need to be explained with solid references.*

Response: Thanks for your good comment. Regarding “**the advanced studies recognized that a large proportion of grain boundaries increase the grain boundary and then bulk resistance**”, the solid references (Ref 1–8) point out that the grain boundary functions differently in microcrystalline and nanocrystalline materials: In microcrystalline materials, grain boundaries do block the carrier flows across grains, increasing the resistance substantially; However, in nanocrystalline materials, grain boundaries act as the highways for carrier migration, decreasing the resistance significantly. The explanations are as follows:

Grain boundaries are regions of higher defect density and disorder with a nanometer-sized width (*J. Phys. C: Solid State Phys.*, 1985, 18, 4079–4119). According to the report by Joachim Maier (Ref

4, *Phys. Chem.*, 1986, 90, 26–33), the transportation of ions or carriers in grain boundary is anisotropic, that is, ions or carriers move more rapidly along grain boundaries (parallel) than across them (perpendicular). It is also well acknowledged that because grain boundaries possess the two key characteristics necessary for enhanced ionic diffusion, *i.e.*, high defect densities (displaced atoms) and high mobilities (interconnected excess free volume), ions or carriers move more rapidly along grain boundaries than across grains or bulk (Ref 2, *Phys. Chem.*, 1984, 88, 1057–1062). For example, Zhu *et al.* employed the conductive atomic force microscopy technique to study the local Li-ion diffusion induced conductance change in Li_xCoO_2 grains, discovering that the grain boundaries have a much lower diffusion energy barrier for Li^+ migrating along (*Sci Rep.*, 2013, 3, 1084).

In microcrystalline materials, the size of grain boundaries is only $\sim 1/1000$ of the grains (given the size of grain is several microns and the width of grain boundary is several nanometers, which are typical values in microcrystalline materials). In this case, ions or carriers mainly transport among grains. When ions migrate from one grain to another, they must move across (perpendicular to) the grain boundaries, which act as barriers due to the anisotropy of ions migration in grain boundaries. Therefore, in microcrystalline materials, grain boundaries increase the overall resistance.

In nanocrystalline materials, the situation is quite different since the width of grain boundaries becomes comparable to the grain size. In such case, carriers mainly transport along (parallel) grain boundaries with enhanced ionic diffusion than the grains (bulk), leading to the significant conductivity enhancement. For example, fast transportation along grain boundaries in nanosized polycrystal silver halides has been reported by Maier (Ref 4, *Phys. Chem.*, 1986, 90, 26). Bellino has also shown that the total ionic conductivity of nanostructured, heavily doped ceria solid electrolytes increases by about one order of magnitude compared with the conductivity of conventional microcrystalline materials (*Adv. Funct. Mater.*, 2006, 16, 107–113).

Consequently, in our work, the self-organized hetero-nanodomains, which compose a considerable number of grain boundaries, can actuate super Li^+ -conduction, leading to a high room temperature σ_{Li^+} of 13.2 mS cm^{-1} .

Reference:

1. *Appl. Phys. Lett.*, 1980, 37, 757–759
2. *Phys. Chem.*, 1984, 88, 1057–1062
3. *Phys. Chem.*, 1985, 89, 355–362

4. *Phys. Chem.*, 1986, 90, 26–33
5. *Phys. Chem.*, 1986, 90, 666–670
6. *J. Chem. Phys.*, 2004, 120, 2375
7. *Nature*, 2000, 408, 946–949
8. *Nat. Mater.*, 2005, 4, 805–815

Comment 2 *The size of self-organized hetero-nanodomains needs to be measured.*

Response: We measured the size of self-organized hetero-nanodomains using Gatan Digital Micrograph software and gave a description in the original manuscript. Please find in **Page 8, line 195**, “As shown in **Figure 2d**, a large amount of nanodomains (the dark area) with a size of 20 – 40 nm are observed, indicating the burst nucleation.”

Comment 3 *PDF of standard Li₃PS₄ needs to be provided in Figure S1a for SXRD patterns. With increasing the nucleation-accelerant "Al₂S₃" phases, e.g., P₂S₇- and P₂S₆- are produced in GCs. Why? The kind of bonding (physical/chemical) that the "Al" has in the crystal structure of Li₃PS₄ is not clear from Raman and 31P MAS NMR spectra. So, XPS/27Al SSNMR needs to be conducted for deep crystal structure analysis. Note! Reference 23 should be removed as it is not published yet. Qu, H. et al. Aluminum Ion Doping Mechanism of Lithium Thiophosphate Solid Electrolytes Revealed with Solid-state NMR. Submitted (2022)*

Response:

1. “PDF of standard Li₃PS₄ needs to be provided in Figure S1a for SXRD patterns.”

It should be noted that the patterns demonstrated in **Figure S1a** in the original manuscript of Li₃PS₄, Li₇P₃S₁₁, Li₄P₂S₆, are the standard JCPDS-ICDD reference phases with the card number of PDF#76-0973, PDF#75-9676 and PDF#08-8948, respectively. AlPS₄ and Al₄(P₂S₆)₃ are obtained from the Fachinformationszentrum ([http://www.fiz-karlsruhe.de/request for de-posed data.html](http://www.fiz-karlsruhe.de/request%20for%20deposited%20data.html)) on quoting the depository numbers CSD-428184 (AlPS₄), CSD-428187 [Al₄(P₂S₆)₃]. **In order not to cause confusion, the PDF card number of Li₃PS₄, Li₇P₃S₁₁, Li₄P₂S₆ are added to Figure S1a in the revised version.**

2. “With increasing the nucleation-accelerant "Al₂S₃" phases, e.g., P₂S₇- and P₂S₆- are produced in GCs. Why?”

We have explained the reasons in the original manuscript, please see **Page 7, line 161-179** for details, or as follows: The solid-state chemistry of the Li_2S - P_2S_5 binary system indicates that the crystalline substances depend on the fraction of Li_2S or P_2S_5 , *i.e.*, the P/S ratio (*J. Mater. Chem. A*, 2017, 5, 18111–18119). For example, when the molar ratio of Li_2S to P_2S_5 is 75 : 25 (P/S = 0.25), pure Li_3PS_4 crystalline can be obtained, within which all sulfur atoms are terminals to build PS_4^{3-} polyhedra. Decreasing the ratio to 70 : 30 (P/S = 0.273) will deposit pure $\text{Li}_7\text{P}_3\text{S}_{11}$ phase composed of PS_4^{3-} isolated and $\text{P}_2\text{S}_7^{4-}$ corner-sharing tetrahedra with a molar ratio of 1 : 1. When Li_2S fraction further decreases to 67 mol% (P/S = 0.333), pure $\text{Li}_4\text{P}_2\text{S}_6$ phase is crystallized, which exclusively has $\text{P}_2\text{S}_6^{4-}$ dumbbells.

In this work, the P/S ratio of all the samples' reagents is fixed to 0.25, where P source is from P_2S_5 and S source is from Li_2S , P_2S_5 and Al_2S_3 . The SXRD and ^{27}Al MAS NMR results (the results of ^{27}Al MAS NMR are discussed in a separate manuscript detailedly, please see the attachment uploaded) suggest Al_2S_3 can participate in solid-state reaction to form Al-doped Li_3PS_4 , *i.e.*, $\text{Li}_{3-3x}\text{Al}_x\text{PS}_4$, while the ^{27}Al MAS NMR results further indicate that the maximum of Al doping content, *i.e.*, the solubility limit of Al, is only 0.06. Below this threshold, all the Al_2S_3 is reactive and thus the P/S ratio is constant 0.25 to generate pure Li_3PS_4 analog, that is, $\text{Li}_{3-3x}\text{Al}_x\text{PS}_4$ with $0 \leq x \leq 0.06$. Beyond the solubility limit, S source from the residual Al_2S_3 is nonreactive and thus the P/S ratio is higher than 0.25. Consequently, with increasing the Al_2S_3 , $\text{P}_2\text{S}_7^{4-}$ and $\text{P}_2\text{S}_6^{4-}$ are produced in GCs.

3. “*The kind of bonding (physical/chemical) that the "Al" has in the crystal structure of Li_3PS_4 is not clear from Raman and 31P MAS NMR spectra. So, XPS/ ^{27}Al SSNMR needs to be conducted for deep crystal structure analysis.*”

According to the phase composition analysis, a Li_3PS_4 analogy phase is identified in LPS7426. Hence, to figure out the local environment of Al in Al-GCs, XPS of the as-prepared LPS7426 is conducted with Al_2S_3 reagent for a fair comparison. As shown in **Figure R7**, for the S 2p spectrum of Al_2S_3 , the peak located at 161.86 eV represents the Al–S species, which is also verified in Al 2p spectrum (74.39 eV) (*Angew. Chem. Int. Ed.*, 2016, 55, 9898–9901). For LPS7426, in addition to the peaks located at 161.78, 162.08 and 163.28 eV representing P–S–Li, P=S, P–S–P, respectively, the peak at 161.92 eV is assigned to Al–S species, which corresponds to the peak located at 77.24 eV in Al 2p spectrum. Consequently, it can be concluded that “Al” has a chemical bonding in the crystal structure of Li_3PS_4 rather than a physical mixture.

Figure R7. S 2p and Al 2p XPS spectra for Al_2S_3 reagent and LPS7426.

The ^{27}Al MAS NMR spectra of Al-GCs have been discussed in detail in a separate manuscript, which was uploaded as an attachment when we submitted our original manuscript. For clarity, the ^{27}Al MAS NMR relevant figures (**Figure 2 – 4**) from the separate manuscript are shown below, while the detailed discussion can be found from Page 6 line 15 to Page 10 line 16 in the attachment (also uploaded in this revision).

Figure 2. (a) ^{27}Al MAS NMR spectra of Al-GCs. (b) ^{27}Al 3QMAS spectrum of LPS7228.

Figure 3. (a) $^{27}\text{Al}\{^{31}\text{P}\}$ REDOR spectra of LPS7030 with increasing rotor periods (T_r). (b) REDOR buildup curve. In $^{27}\text{Al}\{^{31}\text{P}\}$ REDOR experiments, the signal attenuation of the ^{27}Al resonance at 39 ppm is caused by the dipolar recoupling to nearby ^{31}P . (c) ^6Li MAS NMR spectra measured at 153 K and 133 K, for LPS7525 and LPS7426, respectively. (d) Al content in the Al-doped $\text{Li}_{3-3x}\text{Al}_x\text{PS}_4$.

Figure 4. Variable-temperature ^6Li MAS spectra of (a) $\beta\text{-Li}_3\text{PS}_4$ and (b) LPS7228.

4. “Note! Reference 23 should be removed as it is not published yet. Qu, H. et al. Aluminum Ion Doping Mechanism of Lithium Thiophosphate Solid Electrolytes Revealed with Solid-state NMR.

Submitted (2022)

Thanks for your suggestion, we have removed this reference. We will update the references as soon as this separate manuscript is published.

Comment 4 *Figure 2a ³¹P MAS NMR spectra show that impure phase P2S74- and P2S64- are produced with an increase of nucleation-accelerant "Al2S3" and identified for optimized LP7228. However, the emergence of P2S64- in Li3PS4 GC can reduce the σ_{Li^+} to a large extent. For example, how can the authors exclude the impact of impure phases to get a high σ_{Li^+} of 13.2 mS cm⁻¹? Moreover, the Arrhenius plots and the activation energies should be given for related compositions because a more significant ionic conductivity does not always mean smaller activation energy.*

Response: That's a good question. Although Li₄P₂S₆ phase is not indexed from the SXRD spectrum, minor P₂S₆⁴⁻ polyanion is indeed identified from the ³¹P MAS NMR spectrum, mainly due to the evaporation of S during heat treating (*Chem. Mater.*, 2020, 32, 3036–3024; *ACS Appl. Mater. Interfaces*, 2019, 11, 42280–42287). Though P₂S₆⁴⁻ possesses very low conductivity of 10⁻⁷ to 10⁻⁶ S cm⁻¹ at room temperature (*J. Power Sources*, 2006, 159, 193–199; *Solid State Ionics*, 2016, 284, 61–70), its impact on the conductivity of LPS7228 is negligible due to the low content of ~ 3 wt% based on the ³¹P MAS NMR.

Temperature-dependent electrochemical impedance spectroscopy (EIS) are conducted to derive the Arrhenius plots of Al-GCs (**Figure R2a** below). The activation energies (E_a) of Al-GCs are determined based on the Arrhenius equation:

$$\sigma = \frac{\sigma_0}{T} \exp\left(\frac{-E_a}{k_B T}\right) \quad \text{Equation (R1)}$$

where, σ , σ_0 , T , k_B denote the Li⁺ conductivity, exponential prefactor, absolute temperature, Boltzmann constant, respectively.

As shown in **Figure R2b**, a value of 0.310 eV is reached for LPS7525 and decreases to 0.305 eV for LPS7426. With decreasing Li₂S : P₂S₅ ratio, E_a decreases to 0.304 eV for LPS7327, and reaches a minimum of 0.291 eV for LPS7228. The decrease of E_a may be contributed to the formation of Li_{3-3x}Al_xPS₄ with increased carrier's density and hetero-nanodomains. Further reducing the Li₂S : P₂S₅ ratio increases E_a to 0.316 eV (LPS7129), 0.352 eV (LPS7030), 0.375 eV (LPS6931) and 0.398 eV (LPS6535). According to the species evolution of Al-GCs, further decreasing Li₂S content generates

large amount of poor Li⁺ conductor of Li₄P₂S₆ ($E_a = 0.46$ eV, *Chem. Mater.* 2016, 28, 23, 8764–8773) and even non-Li⁺ conductor aluminum thiophosphates, blocking the Li⁺ migration and thus leading to a large E_a .

Figure R2. (a) Arrhenius plots of the conductivity values for Al-GCs. (b) Activation energy for Li⁺ conduction of Al-GCs, as calculated from Arrhenius plots.

Changes in the Supporting Information related to Figure S8 in the revised version: “Temperature-dependent electrochemical impedance spectroscopy (EIS) are conducted to derive the Arrhenius plots of Al-GCs (Figure S8a below). The activation energies (E_a) of Al-GCs are determined based on the Arrhenius equation:

$$\sigma = \frac{\sigma_0}{T} \exp\left(\frac{-E_a}{k_B T}\right) \quad \text{Equation (S1)}$$

where, σ , σ_0 , T , k_B denote the Li⁺ conductivity, exponential prefactor, absolute temperature, Boltzmann constant, respectively.

As shown in Figure S8b, a value of 0.310 eV is reached for LPS7525 and decreases to 0.305 eV for LPS7426. With decreasing Li₂S : P₂S₅ ratio, E_a decreases to 0.304 eV for LPS7327, and reaches a minimum of 0.291 eV for LPS7228. The decrease of E_a may be contributed to the formation of Li_{3-3x}Al_xPS₄ with increased carrier’s density and hetero-nanodomains. Further reducing the Li₂S : P₂S₅ ratio increases E_a to 0.316 eV (LPS7129), 0.352 eV (LPS7030), 0.375 eV (LPS6931) and 0.398 eV (LPS6535). According to the species evolution of Al-GCs, further decreasing Li₂S content generates large amount of poor Li⁺ conductor of Li₄P₂S₆ ($E_a = 0.46$ eV) and even non-Li⁺ conductor aluminum thiophosphates, blocking the Li⁺ migration and thus leading to a large E_a .”

Comment 5 Cyclic voltammetry is an essential measurement to determine the electrochemical voltage window of solid electrolytes. The authors should test cyclic voltammetry by the new method in Adv.

Energy Mater. 2016, 6, 1501590 and then compare the results of LPS7228 to the pristine electrolyte.

Response: Thanks for your suggestion. Using the approach reported in *Adv. Energy Mater.*, 2016, 6, 1501590, we determine the electrochemical voltage window of LPS7228 and pristine electrolyte (LPS7525 in this work). To do so, 30 wt% vapor grown carbon fiber (VGCF) is added into 70 wt % electrolyte to fabricate the electrode composite, LPS7525-VGCF or LPS7228-VGCF. Such composites can increase the contact between electrolyte phase and VGCF, and thus significantly improve the kinetics of the decomposition reaction to obtain the intrinsic electrochemical stability window (EW) of electrolyte. The cyclic voltammetry (CV) scan is conducted in LPS7525-VGCF|Li₆PS₅Cl|Li/In or LPS7228-VGCF|Li₆PS₅Cl|Li/In cells. Specifically, to probe the upper potential of the EW, the cell is cycled at a scan rate of 0.1 mV/s, starting at the open-circuit potential, up to 3.5 V and back to 1 V vs. Li⁺/Li and the cycle ends at the open-circuit potential. To probe the lower potential of the EW, the cell is cycled at a scan rate of -0.1 mV/s, starting at the open-circuit potential, down to 0.5 V and back to 2.0 V vs. Li⁺/Li. As shown in **Figure R8**, the results indicate that the reduction of LPS7525 starts at 2.02 V while the oxidation starts at 2.70 V. For LPS7228, the electrochemical voltage window is 1.53 to 2.94 V, which is wider than that of LPS7525, 2.02 – 2.7 V.

Although LPS7228 has a narrow EW, it can still match high-voltage cathode materials with stable coating. More detailed discussion has been added in the Supporting Information related to **Figure S20** in the revised version.

Figure R8. Cyclic voltammetry profiles of LPS7525-VGCF|Li₆PS₅Cl|Li/In and LPS7228-VGCF|Li₆PS₅Cl|Li/In at a scan rate of 0.1 mV/s in the voltage range of (a) 1.0 – 3.5 V and (b) 0.5 – 2.0 V at room temperature.

Comment 6 *The sulfide SSEs are reported to be reduced at the Li-metal anode side. Does the incorporation of Al₂S₃ prevent the growth of lithium dendrites or suppress the interfacial reaction?*

Response: Yes, the incorporation of Al₂S₃ can both prevent the growth of lithium dendrites and suppress the interfacial reaction, which should be attributed to the *in-situ* formed Li-Al alloy between lithium metal and LPS7228, as discussed in **Page 13 line 306** in the original manuscript. As shown in **Figure S16** in the original manuscript (**Figure S19** in the revised version), XPS analysis of the as-cycled LPS7228/Li interface indicates the *in-situ* formed Li-Al alloy layer on Li metal anode. The positive function of the *in-situ* formed Li-Al layer are:

Firstly, the operating potential of the Li-Al alloy is 0.3 V vs. Li⁺/Li (*J. Power Sources*, 2001, 92, 45–49), which prevents the severe reductive decomposition of the sulfide electrolyte. For example, in Pan's work (*Sci. Adv.*, 2022, 8, eabn4372), a Li-Al alloy anode shows excellent compatibility toward the Li₁₀GeP₂S₁₂ electrolyte, which is widely accepted to be easily reduced when contacts directly with Li metal. In addition, compared with Li metal, the Li-containing alloys such as Li-In, Li-Al, Li-Zn and Li-Sn have higher Li⁺ diffusion coefficients, and better wetting on sulfide electrolyte, which can suppress the growth of Li dendrites (*Chem. Mater.*, 2017, 29, 10, 4181–4189; *Joule*, 2019, 3, 2165; *Adv. Energy Mater.*, 2020, 10, 2000945; *J. Mater. Chem. A*, 2020, 8, 1247–1253).

Comment 7 *The value of critical current density (CCD) has been widely adopted to quantify the dendrite suppression capability of the SSEs. Thus, the CCD of all the composition need to be reported as given in the recently published reference; (*Adv. Funct. Mater.* 2022, 32, 2201528). Also, provide the galvanostatic cycling performance of Li//Li symmetric cells with counterparts.*

Response: Thanks for your good suggestion. Symmetric Li|Al-GCs|Li cells are assembled to test the critical current density (CCD) at room temperature, and the results are shown in **Figure R9**. For the sulfide electrolytes, due to the lithium dendrite growth with increasing current density, the cells show a voltage drop at a certain current value, which is estimated as the CCD. Accordingly, the CCD of Al-GCs from LPS7525 to LPS6535 at room temperature are 0.2, 0.25, 0.5, 0.95, 0.3, 0.2, 0.05 and 0.05 mA cm⁻², respectively.

Figure R9. (a-h) Galvanostatic cycling of Li|Al-GCs|Li symmetric cells with step-increased current densities (0.05 mA cm⁻² for 1 hour) from initial 0.05 mA cm⁻² at room temperature. (i) The CCD of Al-GCs estimated from (a-h).

Then, the galvanostatic cycling performance of Li|Al-GCs|Li cells at 0.2 mA cm⁻²/0.2 mAh cm⁻² and room temperature is depicted in **Figure R10**. A sudden voltage drop appears in the Li|LPS7525|Li symmetric cell after 123 h, illustrating that a short circuit occurs in the cell, while the Li|LPS7426|Li symmetric cell can only deliver stable cycling for 66 h. The Li|LPS7327|Li symmetric cell exhibits a stable cycle for over 160 h at 0.2 mA cm⁻², and it is still cycling steadily. The overpotential of Li|LPS7129|Li symmetric cell shows obvious fluctuations, which could be caused by the interfacial reactions between LPS7129 and Li metal anode (*Adv. Energy Mater.*, 2020, 10, 1903422). Due to the substantial decrease of ionic conductivity, Li|LPS7030|Li can only stable cycle for 18.6 h at 0.2 mA cm⁻². What's worse, due to the lower conductivity of LPS6931 and LPS6535, the Li|LPS6931|Li and Li|LPS6535|Li symmetric cells can not exhibit stable Li stripping/plating at 0.2 mA cm⁻².

Figure R10. Galvanostatic cycling performance of Li|LPS7525|Li, Li|LPS7426|Li, Li|LPS7327|Li, Li|LPS7129|Li, Li|LPS7030|Li, Li|LPS6931|Li and Li|LPS6535|Li at $0.2 \text{ mA cm}^{-2}/0.2 \text{ mAh cm}^{-2}$.

Changes in the Supporting Information related to Figure S14 in the revised version: “Symmetric Li|Al-GCs|Li cells are assembled to test the critical current density (CCD) at room temperature, and the results are shown in **Figure S14**. For the sulfide electrolytes, due to the lithium dendrite growth with increasing current density, the cells show a voltage drop at a certain current value, which is estimated as the CCD. Accordingly, the CCD of Al-GCs from LPS7525 to LPS6535 at room temperature are 0.2, 0.25, 0.5, 0.95, 0.3, 0.2, 0.05 and 0.05 mA cm^{-2} , respectively.”

Comment 8 From *Figure S13*, bulk resistance of Li|LPS7228|Li symmetric cell after 500 cycles is relatively lower (similar to 1st cycle) than after 200 and 1000 cycles, respectively. What happens inside the cell that reduces the bulk resistance of the Li|LPS7228|Li symmetric cell? Explain the reason.

Response: Thanks for your good question. To explain the resistance evolution with cycling, we should first figure out the correspondence between the EIS profile and the resistance of the Li|LPS7228|Li symmetric cell, for example, the bulk resistance, the interfacial resistance, etc. As shown in **Figure R11** (**Figure S16** in the revised version), the EIS plots of the Li|LPS7228|Li symmetrical cell after various cycles show the similar appearance, which composes a suppressed semicircle with a tail connected. According to the previous reports (*Chem. Mater.*, 2016, 28, 2400–2407; *Sci. Adv.*, 2022, 8,

eabn4372), the interfacial resistance of Li-Li symmetric cell is determined by the span of the semicircle and the ohmic (bulk) resistance is determined by the high frequency intercept with the real axis. According to the references (*Chem. Mater.*, 2022, 34, 8, 3659–3669; *ACS Appl. Mater. Interfaces*, 2018, 10, 13588–13597; *Electrochim. Acta*, 2014, 136, 422–429), an equivalent circuit, (R1||CPE1)(R2||CPE2)W (the inset in **Figure R11**), is used to fit the EIS profiles using ZView software (version 2.7) and the fitting results are summarized in **Table R1**. R, CPE, and W in the equivalent circuit represents the resistance, constant phase element, and Warburg impedance, respectively. The ion transportation in the electrolyte is described by R1||CPE1, where R1 represents the bulk resistance; The R2||CPE2 is used to describe the impedance of ion transportation at the interface, where R2 represents the interfacial resistance; W corresponds to the finite-length tail at low frequency (*Adv. Mater.*, 2017, 29, 1605531).

According to the above analysis, the bulk resistances (R1) of Li|LPS7228|Li after the 1st, 200th, 500th, 1000th cycles are 57.5, 58.7, 61.5, 58.5 Ω , respectively. **Hence, the bulk resistance remains nearly unchanged during cycling.** It should be noted that after 200 cycles, the interfacial resistance is almost two times larger than that of the 1st cycle, and then remains stable, which agrees well with the overpotential evolution with cycling (**Figure 4a** in the manuscript).

Figure R11. EIS plots of Li|LPS7228|Li symmetric cell after the 1st, 200th, 500th, 1000th cycles at a current density of 0.2 mA cm⁻² (0.2 or 1 mAh cm⁻²). The equivalent circuit for fitting is shown in the inset. The interfacial resistance (span of the suppressed semicircle) is illustrated by the gray filling as a guide-to-the-eye.

Table R1. EIS fitting parameters for the Li|LPS7228|Li symmetric cell after the 1st, 200th, 500th, 1000th cycles.

	R1 (Ω)	CPE1 (F)	R2 (Ω)	CPE2 (F)
1 st cycle	57.5	3.49×10^{-6}	73.6	1.07×10^{-7}
200 th cycle	58.7	3.21×10^{-6}	130.2	3.13×10^{-7}

500 th cycle	61.5	3.62×10^{-6}	131.8	9.90×10^{-8}
1000 th cycle	58.5	3.93×10^{-6}	133.5	3.68×10^{-7}

Comment 9 Low electronic conductivity is essential to suppress the Li-dendrite formation inside the SSE. Thus, the electronic conductivity of solid electrolytes needs to be measured.

Response: To estimate the electronic conductivity (σ_e), DC polarization of steel|Al-GCs|steel symmetric cells is conducted under a voltage amplitude of 0.1 V. Based on the polarization curves in **Figure R12a**, σ_e of Al-GCs (from LPS7525 to LPS6535) are calculated to be 4.74, 4.83, 5.76, 4.40, 5.67, 3.84, 4.76, 5.46×10^{-9} S cm⁻¹ (**Figure R12b**), respectively, which are comparable to that of previously reported GCs (*Adv. Mater.*, 2021, 2006577).

Figure R12. (a) DC polarization curves of steel|Al-GCs|steel symmetric cells using a voltage amplitude of 0.1 V. (b) The estimated σ_e of various Al-GCs at room temperature.

Changes in the Supporting Information related to Figure S9 in the revised version: “Low electronic conductivity (σ_e) is important for solid electrolytes, in order to minimize self-discharge, and reduce dendritic lithium formation. The DC polarization curves of Al-GCs under an applied voltage of 0.1 V are shown in **Figure S9a**, based on which the σ_e of Al-GCs (from LPS7525 to LPS6535) are calculated to be 4.74, 4.83, 5.76, 4.40, 5.67, 3.84, 4.76, 5.46×10^{-9} S cm⁻¹ (**Figure S9b**), respectively,

which are similar to that of reported GCs.”

Changes in the revised manuscript, Page 23, line 550-553 insert: “DC polarization measurement is carried out to determine the electronic conductivity. Subsequently, time dependence of the current is measured under constant voltage 0.1 V for 2 h by using the electrochemical working station Biologic VMP-300.”

Comment 10 It is essential to provide more basic electrochemical properties of LCO/LPS7228/Li battery, e.g., cyclic voltammetry, long cycling performance under high current density, e.g., 0.5 C and EIS before and after measurements.

Response: Thanks for your suggestions. **Figure R13a** shows the cyclic voltammetry (CV) of the LiNbO₃ coated LiCoO₂|LPS7228|Li at a scan rate of 0.1 mV s⁻¹, which exhibits well-defined redox peaks, corresponding to the main lithiation/delithiation process. The symmetry of oxidative/reductive peaks indicates a good reversibility of the intercalation/deintercalation process.

According to your suggestion, the cycling performance of LiCoO₂|LPS7228|Li ASSB at 0.5 C, room temperature is evaluated. As shown in **Figure R13b** and **R13c**, a reversible specific discharge capacity of 120.5 mAh g⁻¹ is achieved at 0.5 C for the first cycle in the voltage range of 3.0 – 4.3 V with a high CE of 96.1 %. A reversible specific capacity of 103.7 mAh g⁻¹ is maintained after 110 cycles, showing a good retention of 86.1 % with an average CE of 99.2 %, suggesting highly reversible Li⁺ intercalation/deintercalation. To understand the probable reasons for capacity fade, EIS measurements on the LiNbO₃ coated LiCoO₂|LPS7228|Li are conducted. **Figure R13d** shows the Nyquist plots of the ASSB after the first and 110th cycle. The ohmic resistances remain stable, while the interfacial resistances increase, accounting for the capacity fade with cycling.

Figure R13. (a) CV profile of LiCoO₂|LPS7228|Li. (b) Cycle behavior and (c) charge-discharge curves of LiCoO₂|LPS7228|Li ASSB at 0.5 C and room temperature. (d) EIS plots of LiCoO₂|LPS7228|Li ASSB after the 1st and 110th discharging.

Changes in the Supporting Information related to Figure S22 in the revised version: “Figure S22a shows the cyclic voltammetry (CV) of the LiNbO₃ coated LiCoO₂|LPS7228|Li at a scan rate of 0.1 mV s⁻¹, which exhibits well-defined redox peaks, corresponding to the main lithiation/delithiation process. The symmetry of oxidative/reductive peaks indicates a good reversibility of the intercalation/deintercalation process. As shown in Figure S22b and S22c, the cycling performance of LiCoO₂|LPS7228|Li ASSB at 0.5 C is evaluated. A reversible specific discharge capacity of 120.5 mAh g⁻¹ is achieved at 0.5 C for the first cycle in the voltage range of 3.0 – 4.3 V with a high CE of 96.1 %. A reversible specific capacity of 103.7 mAh g⁻¹ is maintained after 110 cycles, showing a good retention of 86.1 % with an average CE of 99.2 %, suggesting highly reversible Li⁺ intercalation/deintercalation. To understand the probable reasons for capacity fade, EIS measurements on the LiNbO₃ coated LiCoO₂|LPS7228|Li are conducted. Figure S22d shows the Nyquist plots of the ASSB after the first and 110th cycle. The ohmic resistances remain stable, while the interfacial resistances increase, accounting for the capacity fade with cycling.”

Reviewer #3:

General Comments: *Today it is widely acknowledged that the low ionic conductivity of solid electrolytes along with its interfacial instability are the main obstacles in safer solid state battery development. The manuscript shows the way of increasing Li₂S-P₂S₅ glass-ceramics ionic conductivity by tuning the microstructure. The authors use the addition of various sulphides as a tool for controlling the composition and microstructure of the obtained solid electrolytes. Unlike many other papers in the field, here author show the clear idea rather than going through simple trial-and-error route. I think the report is suitable for publishing in Nature communications, however, some point should be addressed by the authors before acceptance of the manuscript.*

Response: We thank the reviewer for the strong endorsement of our work.

Comment 1 *The authors claim that the interface between lithium and glass-ceramics with Al₂S₃ added and the optimal composition stays smooth during Li plating/stripping. First, it is not clear what happens with the interface when other sulfide modifiers are used. Second, I'm wondering whether the interface with argyrodite can be stable if Li-Al alloy will be used instead of pure Li. Is there any difference between in situ formed Li-Al layer and Li-Al alloy used as anode? To the best of my knowledge, Li-Al alloy foils undergo severe dendrite formation during cycling as well. Third, I think the XPS data in Fig. S16 and its discussion are not convincing. I found no experimental details on XPS measurements which are very important as the electrolyte sample are electronic insulators thus interpretation of the spectra can be strongly affected by charging and differential charring effects.*

Response:

1. “First, it is not clear what happens with the interface when other sulfide modifiers are used.”

To figure out what happens with the interface when other sulfide modifiers are used, symmetric Li-Li cells using Si-LPS7327 and Ga-LPS7228 GC electrolytes, due to the highest ionic conductivity among their respective components, are assembled and the Li stripping/plating behaviour is tested at a current density of 0.2 mA cm⁻² (0.2 mAh cm⁻²). As shown in **Figure R14a**, Li|Si-LPS7327|Li exhibits a continuously increasing overpotential from 32 to 521 mV within 135 h, corresponding to a significant increase of the interfacial resistance from 118 Ω to 1925.8 Ω (**Figure R14b** and **R14c**).

To explain the interface instability, XPS is used to obtain complementary chemical insight

into the Li|Si-LPS7327 interface. As shown in **Figure R14e**, the S 2p spectrum of the pristine Si-LPS7327 sample shows peaks assigned to P–S–Li, P=S, P–S–P binding at 161.87, 162.08, 163.14 eV, respectively, corresponding to 132.22 (PS_4^{3-}) and 132.53 ($\text{P}_2\text{S}_7^{4-}$) eV in the P 2p spectrum (*Chem. Mater.*, 2019, 31, 3745–3755; *Chem. Mater.*, 2020, 32, 6123–6136). In addition, the peaks at 161.82 eV in S 2p spectrum and 101.40 eV in Si 2p spectrum can be attributed to SiS_4^{4-} anions (*Chem. Mater.*, 2022, 34, 8, 3659–3669). After cycling, severe reduction of Si-LPS7327 is detected at the Li/Si-LPS7327 interface. Specifically, large amount of Li_2S can be detected at 159.88 eV in the S 2p spectrum, while the presence of reduced P species in the spectrum of Li|Si-LPS7327 also can be detected, indicating the reduction of P_xS_y polyanions. In addition, impurities of Si^0 (99.66 eV) and Li-Si alloy (97.76 eV) can be detected from the Li|Si-LPS7327 interface, suggesting the reduction of SiS_4^{4-} (*Surf. Sci. Spectra*, 2013, 20, 36–42; *Chem. Mater.*, 2012, 24, 1107–1115). Another impurity at 103.07 eV is caused by SiO_2 , which could be due to the side reactions with oxygen during the sample transfer to the chamber. Consequently, at the Li/Si-LPS7327 interface, formation of large amount of poorly conductive Li_2S and Si^0 , which also undergoes severe volume change during lithiation/delithiation to result in contact loss (*ACS Nano*, 2014, 8, 8591 – 8599; *J. Power Sources*, 2007, 163, 1003–1039), accounts for the continuous overpotential increase with Li plating/stripping.

The X-ray microtomography is also conducted to observe the Li/Si-LPS7327 interface evolution with cycling. **Figure R14g** shows the sliced images of the reconstructed tomographic volumes of pristine and as-cycled Li|Si-LPS7327|Li. A spiky interface and cracks (marked as yellow arrows) are clearly observed. What's worse, the Si-LPS7327 particles at the interface have been seriously pulverized (marked as yellow dash line). All the experimental analysis demonstrates the unstable interface between SiS_2 tuned GCs and lithium metal.

In contrast, the Li|Ga-LPS7228|Li symmetric cell shows an initial overpotential of 15.8 mV, which remains stable for 135 h, in line with the nearly unchanged interfacial resistance (**Figure R14d**). **Figure R14f** shows the Ga 2p_{3/2}, S 2p, and P 2p XPS spectra for pristine Ga-LPS7228 and the as-cycled Ga-LPS7228/Li interface. The emerging signals at 1119.6 and 1117.8 eV of Ga 2p can be attributed to the formation of Li-Ga alloy and Gallium (*J. Electrochem. Soc.*, 2016, 163, A2488–A2493), indicating the reduction of Ga^{3+} to lower-valency Ga species, while no other

obvious reductive species, *e.g.*, Li_2S , are distinguished from P 2p or S 2p spectra. It should be noted that trace amounts of residual liquid metal Ga can relieve the stress generated from Li plating, while the solidified Li-Ga alloy with a low surface ion diffusion barrier can guarantee a rapid and homogeneous Li deposition (*J. Mater. Chem. A*, 2020, 8, 17415–17419; *Nat. Energy*, 2018, 3, 227–235), which is verified by the smooth and integrated interface between Ga-LPS7228 and Li metal anode after cycling as observed in **Figure R14h**. All the experimental analysis demonstrates the stable interface between Ga_2S_3 tuned GCs and lithium metal.

Figure R14. (a) Galvanostatic polarization of the Li|Si-LPS7327|Li and Li|Ga-LPS7228|Li cells at a current of 0.2 mA cm^{-2} and areal specific capacity of 0.2 mAh cm^{-2} at room temperature. EIS plots of Li|Si-LPS7327|Li symmetric cell (b) before and (c) after cycling. (d) EIS plots of Li|Ga-LPS7228|Li symmetric cell before and after cycling. (e) S 2p, P 2p, and Si 2p XPS spectra for pristine Si-LPS7327 and the as-cycled Si-LPS7327/Li interface. (f) Ga 2p_{3/2}, S 2p, and P 2p XPS spectra for pristine Ga-LPS7228 and the as-cycled Ga-LPS7228/Li interface. Sliced images of the reconstructed tomographic volumes of pristine and as-cycled (g) Li|Si-LPS7327|Li and (h) Li|Ga-LPS7228|Li symmetric cells.

2. “Second, I’m wondering whether the interface with argyrodite can be stable if Li-Al alloy will be

used instead of pure Li. Is there any difference between in situ formed Li-Al layer and Li-Al alloy used as anode? To the best of my knowledge, Li-Al alloy foils undergo severe dendrite formation during cycling as well.”

Thanks for your good question. To evaluate the compatibility of Li-Al alloy anode toward the $\text{Li}_6\text{PS}_5\text{Cl}$ electrolyte, two different Li-Al alloys, *i.e.*, Li-Al alloy 1# (molar ratio of Li : Al = 1 : 1) and Li-Al alloy 2# (molar ratio of Li : Al = 2.25 : 1) are prepared according to Ref. *J. Mater. Chem. A*, 2019, 7, 25415–25422. The preparation processes are as follows: Al foil is first sanded to remove the oxide layer from the surface (**Figure R15a**); then the Li-Al alloy is obtained by attaching the preprocessed Al foil to the Li foil at a specific stoichiometric ratio; After rested under 300 MPa for 12 h, Li foil and Al foil spontaneously alloy, as indicated by the XRD patterns in **Figure R15e**.

Symmetric Li-Al alloy 1#| $\text{Li}_6\text{PS}_5\text{Cl}$ |Li-Al alloy 1# and Li-Al alloy 2#| $\text{Li}_6\text{PS}_5\text{Cl}$ |Li-Al alloy 2# cells are assembled and galvanostatic Li stripping/plating tests are conducted to assess the stability of Li-Al alloy/ $\text{Li}_6\text{PS}_5\text{Cl}$ interface. As shown in **Figure R15f**, the Li-Al alloy 1#| $\text{Li}_6\text{PS}_5\text{Cl}$ |Li-Al alloy 1# symmetric cell exhibits an overpotential of ~ 38 mV, which maintains stable for over 200 h. On the contrary, the Li-Al alloy 2#| $\text{Li}_6\text{PS}_5\text{Cl}$ |Li-Al alloy 2# symmetric cell survives only 24.8 h with a short circuit appearing afterwards. Then, the cells are imaged using X-ray microtomography and the corresponding slices through typical tomograms are shown in **Figure R15g** and **R15h**, respectively. The Li-Al alloy 1#/ $\text{Li}_6\text{PS}_5\text{Cl}$ interfaces are devoid of any noticeable features, and no dendritic structures are observed (**Figure R15g**). In comparison, the interface of Li-Al alloy 2#/ $\text{Li}_6\text{PS}_5\text{Cl}$ becomes spiny. What’s worse, dendritic structures are clearly seen in **Figure R15h** marked by yellow arrows.

According to the above results and previous reports (*Sci. Adv.*, 2022, 8, eabn4372), whether the Li-Al/argyrodite interface is stable depends on the molar ratio of Li : Al (or the Li proportion) in Li-Al alloy. In terms of the physical/chemical properties, the Li-rich Li-Al alloy 2# is closer to the pure Li metal and hence the Li-rich Li-Al alloy 2# has a lower electrode potential than that of Li-poor Li-Al alloy 1# (*i.e.*, 0.23 vs. 0.38 V vs. Li^+/Li , respectively) (*Energy Storage Materials*, 2020, 25, 93–99). Thus, $\text{Li}_6\text{PS}_5\text{Cl}$ electrolytes are prone to be chemically reduced by Li-Al alloy 2#, leading to uneven Li deposition thus further resulting in the formation of Li dendrites to cause short-circuits eventually. Consequently, $\text{Li}_6\text{PS}_5\text{Cl}$ is not compatible with the Li-Al alloy with

lower Al proportion, and the Li-Al alloy with lower Al proportion still undergoes severe dendrite formation during cycling, while the Li-Al alloy with higher Al proportion does not.

As for the question “Is there any difference between *in situ* formed Li-Al layer and Li-Al alloy used as anode?” Yes, we think the *in-situ* formed Li-Al alloy layer functions quite differently than the Li-Al alloy used as anode.

For Li-Al alloy anode, volume swells/contracts concomitantly with lithiation/delithiation, which can lead to serious contact loss during cycling (*Nat. Energy*, 2017, 2, 17119). In comparison, because the *in-situ* formed Li-Al alloy is line phase, *i.e.*, not solid solution, it is compositionally dynamic balance on cycling by virtue of contact with the lithium foil. Hence, it does not undergo severe volume changes. In addition, as indicated by the atomic force microscopy (AFM) tests, Li metal is softer than Li-Al alloy (Li : Al, 1 : 1). Consequently, it can be speculated that Li metal, and hence the *in-situ* formed Li-Al alloy layer, contacts more closely with the electrolyte than Li-Al alloy anode does. The close contact will uniformize the Li⁺ flux during the Li plating/stripping processes, which can suppress the formation of Li dendrites. Consequently, the *in-situ* formed Li-Al alloy layer functions quite differently compared to the Li-Al alloy used as anode.

The detailed illustration of the AFM measurement in Figure R15i-k: Firstly, the AFM tip is brought to approach to the metal surface. An attraction force is received for the tip as it approaches to the surface infinitely, resulting in the cantilever getting bent. Further approach of the AFM tip towards the surface of metal surface causes loading of the tip onto the metal surface. The cantilever is slowly retracted from the surface while the tip is still in contact with the metal surface until it completely separates from the metal surface. **Figure R15j and R15k** shows the force-distance curves of pure Li film and Li-Al alloy 1#, respectively. According to previous reports (*Membranes*, 2012, 2, 783–803; *J. Colloid Interface Sci.*, 1975, 53, 314–326), the slope of the force-distance curve measured during the loading step that can be used to qualitatively describe the hardness of the measured sample. Clearly, Li-Al alloy 1# has a higher slope's value than that of pure Li metal, indicating that lithium metal is softer than Li-Al alloy.

Figure R15. Digital images of (a) Al foil, (b) the Al foil attaching to the Li foil with stoichiometric ratio, and (c-d) pressed under 300 MPa for 12 h. (e) XRD patterns of the as-prepared Li-Al alloys. (f) Galvanostatic Li stripping/plating profiles of the Li-Al alloy/ $\text{Li}_6\text{PS}_5\text{Cl}$ /Li-Al alloy at 0.2 mA cm^{-2} and 0.2 mAh cm^{-2} . Sliced images of the reconstructed tomographic volumes of (g) Li-Al alloy 1#/ $\text{Li}_6\text{PS}_5\text{Cl}$ /Li-Al alloy 1# and (h) Li-Al alloy 2#/ $\text{Li}_6\text{PS}_5\text{Cl}$ /Li-Al alloy 2#. (i) Schematic diagram illustrating the working principle of the AFM measurement. The force-distance curves of (j) pure Li film and (k) Li-Al alloy 1#.

3. “Third, I think the XPS data in Fig. S16 and its discussion are not convincing. I found no experimental details on XPS measurements which are very important as the electrolyte sample are electronic insulators thus interpretation of the spectra can be strongly affected by charging and differential charring effects.”

Thanks for your comment. We are sorry that we omitted the experimental details of XPS in the original manuscript. We have added this in our revised manuscript.

Changes in the manuscript, Page 25, line 597-606: “The samples are placed onto double-sided tape fixed to clean glass slides and placed in a vacuum transfer holder inside an Ar-filled glovebox.

An X-ray photoelectron spectroscopy spectrometer (Thermo Scientific Model K-Alpha XPS) with a monochromatized Al K α source (1486.6 eV) is used to obtain the surface chemistry of the samples. A 400 μm X-ray spot size is used to maximize the signal intensity and to obtain an average surface composition over a large area. The sample surface is cleaned via Ar⁺ sputtering with an acceleration voltage of 0.5 kV for 200 s. A dual beam charge neutralization is applied for charge compensation. The base pressure in the analysis chamber is 3×10^{-10} mbar. The pass energy is 23.5 eV. Spectra are charge corrected using the C 1s core level peak set to 284.8 eV. Data evaluation is performed with the software Thermo Avantage XPS software package (version 5.976).”

Comment 2 *The understanding of the phase composition and evolution in the system is of a high importance. The authors put a lot of efforts in the XRD analysis. However, there are no details on Rietveld refinements given in the Supplementary info - no structural models, no refinement algorithm (sequence for refining different parameters), etc.*

Response: Thanks for your suggestions. Rietveld refinements were performed using GSAS II program. Initially, the structural information obtained from the Materials Project database (*Sci Data*, 2015, 2, 150009). These steps were used during the refinements: (i) scale factor, (ii) 12 coefficients for a Chebyshev function background, (iii) peak shape parameters, (iv) lattice parameters, (v) zero error, (vi) fractional atomic coordinates, (vii) atomic occupancies. Then, multiple correlated parameters were refined simultaneously, and the related constraints Tables can be found separately in Supporting Information.

Changes in the manuscript, Page 24, line 575-580: “Initially, the structural information obtained from the Materials Project database. These steps are used during the refinements: (i) scale factor, (ii) 12 coefficients for a Chebyshev function background, (iii) peak shape parameters, (iv) lattice parameters, (v) zero error, (vi) fractional atomic coordinates, (vii) atomic occupancies. Then, multiple correlated parameters are refined simultaneously, and the related constraints Tables can be found separately in Supporting Information.”

Changes in the Supporting Information:

Table S2-1. Crystallographic data of LPS7525. The lattice parameters, fractional atomic coordinates, occupancies are obtained from the Rietveld refinements against the SXRD data.

Li₃PS₄ (space group *P n m a*)

$\lambda = 0.82657 \text{ \AA}$ lattice parameter $a = 12.8846(1) \text{ \AA}$, $b = 8.1171(4) \text{ \AA}$, $c = 6.1296(5) \text{ \AA}$
 $R_{wp} = 8.625 \%$, Goodness of fit = 2.73

Atom	Wyckoff position	Atomic coordinates			Occ.
		x	y	z	
S1	8d	0.1513(3)	0.0470(6)	0.2791(3)	1
S2	4c	0.9369(7)	0.25	0.2570(4)	1
S3	4c	0.0999(4)	0.25	0.8307(1)	1
P	4c	0.0860(6)	0.25000	0.1720(7)	1
Li1	8d	0.3180(1)	0.0170(1)	0.1390(1)	1
Li2	4b	0	0	0.5	0.7048(8)
Li3	4c	0.4420(1)	0.25	0.8307(1)	0.2951(2)

Table S2-2. Crystallographic data of LPS7426. The lattice parameters, fractional atomic coordinates, occupancies are obtained from the Rietveld refinements against the SXRD data.

Li₃PS₄ (space group *P n m a*)
 $\lambda = 0.82657 \text{ \AA}$ lattice parameter $a = 13.0206(5) \text{ \AA}$, $b = 8.2041(1) \text{ \AA}$, $c = 6.1894(2) \text{ \AA}$
 $R_{wp} = 6.298 \%$, Goodness of fit = 1.90

Atom	Wyckoff position	Atomic coordinates			Occ.
		x	y	z	
S1	8d	0.1498(2)	0.0518(4)	0.2152(3)	1
S2	4c	0.4402(4)	0.25	0.2585(9)	1
S3	4c	0.1018(7)	0.25	0.6648(7)	1
P	4c	0.0853(1)	0.25000	0.3254(9)	1
Li1	8d	0.3213(1)	0.0245(1)	0.1177(7)	1
Li2	4b	0	0	0.5	0.6822(8)
Li3	4c	0.4990(1)	0.25	0.8890(1)	0.3177(2)

Table S2-3. Crystallographic data of LPS7327. The lattice parameters, fractional atomic coordinates, occupancies are obtained from the Rietveld refinements against the SXRD data.

Li₃PS₄ (space group *P n m a*)
 $\lambda = 0.82657 \text{ \AA}$ lattice parameter $a = 12.8802(7) \text{ \AA}$, $b = 8.1437(5) \text{ \AA}$, $c = 6.1421(2) \text{ \AA}$
 $R_{wp} = 5.973 \%$, Goodness of fit = 1.80
Phase fraction : 93.5 wt%

Atom	Wyckoff position	Atomic coordinates			Occ.
		x	y	z	
S1	8d	0.1513(3)	0.0470(6)	0.2791(3)	1
S2	4c	0.9369(7)	0.25	0.2570(4)	1
S3	4c	0.0999(4)	0.25	0.8307(1)	1
P	4c	0.0860(6)	0.25000	0.1720(7)	1
Li1	8d	0.3180(1)	0.0170(1)	0.1390(1)	1
Li2	4b	0	0	0.5	0.7048(8)
Li3	4c	0.4420(1)	0.25	0.8307(1)	0.2951(2)

Li₇P₃S₁₁ (space group *P*−*I*)
 $\lambda = 0.82657 \text{ \AA}$ lattice parameter $a = 6.1141(9) \text{ \AA}$, $b = 12.4635(4) \text{ \AA}$, $c = 12.6377(6) \text{ \AA}$
 $R_{wp} = 5.973 \%$, Goodness of fit = 1.80
Phase fraction : 6.5 wt%

Atom	Wyckoff position	Atomic coordinates			Occ.
		x	y	z	
S1	2i	0.1019(7)	0.8669(3)	0.1146(1)	1
S2	2i	0.0920(5)	0.1834(9)	0.1869(1)	1
S3	2i	0.4623(7)	0.1396(8)	-0.0018(8)	1
S4	2i	0.2118(2)	0.3542(5)	0.0035(8)	1
S5	2i	0.4126(9)	0.6726(1)	0.2067(5)	1
S6	2i	0.3181(1)	0.4408(8)	0.3177(6)	1
S7	2i	0.7734(2)	0.4674(8)	0.1983(3)	1
S8	2i	0.1767(5)	0.8045(8)	0.3799(2)	1
S9	2i	0.2804(6)	0.1732(1)	0.4741(9)	1
S10	2i	0.1680(5)	0.6575(6)	0.5529(8)	1
S11	2i	0.6877(1)	0.0058(6)	0.3092(3)	1
P1	2i	0.1431(5)	0.1858(7)	0.0025(5)	1
P2	2i	0.4724(1)	0.4584(8)	0.2200(4)	1
P3	2i	0.8825(1)	0.1465(7)	0.4254(6)	1
Li1	2i	-0.2043(1)	0.2175(4)	0.4067(6)	1
Li2	2i	-0.1407(2)	0.4778(2)	0.2925(9)	1
Li3	2i	0.0049(8)	0.4307(5)	0.4836(7)	1
Li4	2i	0.2868(3)	-0.2560(8)	0.0609(1)	1
Li5	2i	0.2439(6)	0.6322(3)	0.2960(8)	1
Li6	2i	0.4387(9)	0.3481(1)	0.2757(1)	1
Li7	2i	0.1270(2)	-0.0107(5)	0.7409(8)	1

Table S2-4. Crystallographic data of LPS7228. The lattice parameters, fractional atomic coordinates, occupancies are obtained from the Rietveld refinements against the SXRD data.

Li₃PS₄ (space group *P n m a*)
 $\lambda = 0.82657 \text{ \AA}$ lattice parameter $a = 12.8802(7) \text{ \AA}$, $b = 8.1437(5) \text{ \AA}$, $c = 6.1421(2) \text{ \AA}$
 $R_{wp} = 7.747 \%$, Goodness of fit = 2.42
Phase fraction : 27.2 wt%

Atom	Wyckoff position	Atomic coordinates			Occ.
		x	y	z	
S1	8d	0.1540(2)	0.0471(3)	0.2158(9)	1
S2	4c	0.4351(9)	0.25	0.2644(6)	1
S3	4c	0.0929(9)	0.25	0.6682(1)	1
P	4c	0.0879(3)	0.25	0.3109(9)	1
Li1	8d	0.3464(1)	-0.0701(8)	0.0664(4)	1

Li2	4b	0	0	0	0.6930(8)
Li3	4c	0.7980(5)	0.25	0.8182(7)	0.3069(2)

Li₇P₃S₁₁ (space group *P-1*)

$\lambda = 1.54051 \text{ \AA}$ lattice parameter a = 6.0737(2) \AA , b = 12.3049(1) \AA , c = 12.5403(9) \AA

$R_{wp} = 7.747 \%$, Goodness of fit = 2.42

Phase fraction : 72.8 wt%

Atom	Wyckoff position	Atomic coordinates			Occ.
		x	y	z	
S1	2i	0.0866(5)	0.8444(6)	0.1158(4)	1
S2	2i	0.1220(1)	0.1599(6)	0.1712(1)	1
S3	2i	0.4250(1)	0.1329(1)	0.0358(3)	1
S4	2i	0.2198(3)	0.3824(9)	0.0261(1)	1
S5	2i	0.4354(9)	0.6775(8)	0.2002(5)	1
S6	2i	0.3228(1)	0.4659(8)	0.3273(9)	1
S7	2i	0.8264(6)	0.4753(7)	0.1904(1)	1
S8	2i	0.1478(1)	0.8176(1)	0.3938(6)	1
S9	2i	0.2842(8)	0.1682(6)	0.4641(6)	1
S10	2i	0.1490(4)	0.6701(1)	0.5479(7)	1
S11	2i	0.6966(4)	0.0249(3)	0.3115(6)	1
P1	2i	0.1207(1)	0.2193(8)	0.0406(8)	1
P2	2i	0.5231(4)	0.5124(3)	0.2442(9)	1
P3	2i	0.8583(2)	0.1252(1)	0.4184(7)	1
Li1	2i	0.1011(3)	0.2867(2)	0.4506(3)	1
Li2	2i	-0.0237(5)	0.5789(9)	0.2884(3)	1
Li3	2i	0.7264(8)	0.1511(1)	0.1038(3)	1
Li4	2i	0.2938(5)	-0.0267(7)	0.2780(4)	1
Li5	2i	0.2701(4)	0.5245(8)	0.1077(1)	1
Li6	2i	0.4328(4)	0.3646(2)	-0.0420(3)	1
Li7	2i	0.2332(3)	0.0389(2)	0.3858(2)	1

Table S2-5. Crystallographic data of LPS7129. The lattice parameters, fractional atomic coordinates, occupancies are obtained from the Rietveld refinements against the SXRD data.

Li₃PS₄ (space group *Pnm*)

$\lambda = 0.82657 \text{ \AA}$ lattice parameter a = 12.8644(1) \AA , b = 8.1253(5) \AA , c = 6.1223(1) \AA

$R_{wp} = 4.787 \%$, Goodness of fit = 1.48

Phase fraction : 24.7 wt%

Atom	Wyckoff position	Atomic coordinates			Occ.
		x	y	z	
S1	8d	0.1551(1)	0.0458(3)	0.2148(2)	1
S2	4c	0.4346(7)	0.25	0.2574(3)	1
S3	4c	0.1068(1)	0.25	0.6634(9)	1

P	4c	0.0872(4)	0.25	0.3281(7)	1
Li1	8d	0.3562(2)	0.0130(4)	0.0610(5)	1
Li2	4b	0	0	0.5	0.6827(4)
Li3	4c	0.4990(2)	0.25	0.8890(3)	0.3175(6)

Li₇P₃S₁₁ (space group *P-1*)

$\lambda = 0.82657 \text{ \AA}$ lattice parameter $a = 6.0737(2) \text{ \AA}$, $b = 12.3049(1) \text{ \AA}$, $c = 12.5403(9) \text{ \AA}$

$R_{wp} = 4.787 \%$, Goodness of fit = 1.48

Phase fraction : 47.9 wt%

Atom	Wyckoff position	Atomic coordinates			Occ.
		x	y	z	
S1	2i	0.0969(3)	0.8606(8)	0.1195(3)	1
S2	2i	0.0713(8)	0.1765(7)	0.1658(8)	1
S3	2i	0.4771(1)	0.1600(4)	0.0320(6)	1
S4	2i	0.2227(5)	0.3824(3)	0.0471(4)	1
S5	2i	0.4427(4)	0.6694(3)	0.2116(3)	1
S6	2i	0.4119(6)	0.4914(3)	0.3493(6)	1
S7	2i	0.7905(3)	0.4906(2)	0.1926(3)	1
S8	2i	0.1708(6)	0.8406(4)	0.4033(4)	1
S9	2i	0.2455(7)	0.1721(1)	0.4538(7)	1
S10	2i	0.1655(1)	0.6781(5)	0.5640(1)	1
S11	2i	0.6719(6)	0.0228(9)	0.3124(1)	1
P1	2i	0.1728(8)	0.2062(6)	0.0346(7)	1
P2	2i	0.4691(2)	0.5071(2)	0.2034(6)	1
P3	2i	0.8902(7)	0.1600(5)	0.4415(5)	1
Li1	2i	0.1988(1)	0.3310(3)	0.4080(6)	1
Li2	2i	0.0401(2)	0.6381(7)	0.3657(7)	1
Li3	2i	0.7460(1)	0.2160(1)	0.2400(1)	1
Li4	2i	0.2720(2)	0.0580(1)	0.2300(9)	1
Li5	2i	0.0501(9)	0.6410(1)	0.1632(4)	1
Li6	2i	0.6290(1)	0.3630(1)	0.0020(7)	1
Li7	2i	0.4330(6)	0.1390(1)	0.6340(1)	1

Li₄P₂S₆ (space group *P 63/m c m*)

$\lambda = 0.82657 \text{ \AA}$ lattice parameter $a = 6.0703(8) \text{ \AA}$, $c = 6.1223(1) \text{ \AA}$

$R_{wp} = 4.787 \%$, Goodness of fit = 1.48

Phase fraction : 27.4 wt%

Atom	Wyckoff position	Atomic coordinates			Occ.
		x	y	z	
S	6g	0.3237(2)	0	0.25	1
P	4e	0	0	0.1715(1)	0.5
Li	4d	0.3333(3)	0.6666(7)	0	1

Table S2-6. Crystallographic data of the as-prepared AIPS₄. The lattice parameters, fractional atomic coordinates, occupancies are obtained from the Rietveld refinements against the XRD data.

AIPS ₄ (space group $P-42c$)					
$\lambda = 1.54051 \text{ \AA}$ lattice parameter $a = 5.6825(1) \text{ \AA}$, $c = 9.1103(8) \text{ \AA}$					
$R_{wp} = 3.916 \%$, Goodness of fit = 2.28					
Impurity phase: 11.2 wt% of Al ₄ (P ₂ S ₆) ₃					
Atom	Wyckoff position	Atomic coordinates			Occ.
		x	y	z	
S	8n	0.2105(5)	0.2786(8)	0.1231(4)	1
P	2d	0	0.5	0.25	1
Al	2a	0	0	0.25	1

Table S2-7. Crystallographic data of the as-prepared Li_{2.82}Al_{0.06}PS₄. The lattice parameters, fractional atomic coordinates, occupancies are obtained from the Rietveld refinements against the SXRD data.

Li _{2.82} Al _{0.057} PS ₄ (space group $Pnm a$)					
$\lambda = 1.54051 \text{ \AA}$ lattice parameter $a = 12.9057(1) \text{ \AA}$, $b = 8.1417(3) \text{ \AA}$, $c = 6.1369(2) \text{ \AA}$					
$R_{wp} = 8.265\%$, Goodness of fit = 2.73					
Atom	Wyckoff position	Atomic coordinates			Occ.
		x	y	z	
S1	8d	0.1566(6)	0.0388(8)	0.2812(7)	1
S2	4c	0.9300(3)	0.25	0.2617(8)	1
S3	4c	0.1022(8)	0.25	0.8051(8)	1
P	4c	0.0859(4)	0.25	0.1722(6)	1
Al	4b	0	0	0.5	0.2292(8)
Li1	8d	0.3452(1)	0.9477(1)	0.4499(9)	1
Li2	4b	0	0	0.5	0.5414(9)
Li3	4c	0.9960(1)	0.25	0.8890(1)	0.2865(5)

Table S2-8. Crystallographic data of the as-prepared Li₇P₃S₁₁. The lattice parameters, fractional atomic coordinates, occupancies are obtained from the Rietveld refinements against the XRD data.

Li ₇ P ₃ S ₁₁ (space group $P-1$)					
$\lambda = 1.54051 \text{ \AA}$ lattice parameter $a = 6.0737(2) \text{ \AA}$, $b = 12.3049(1) \text{ \AA}$, $c = 12.5403(9) \text{ \AA}$					
$R_{wp} = 2.743 \%$, Goodness of fit = 1.05					
Atom	Wyckoff position	Atomic coordinates			Occ.
		x	y	z	
S1	2i	0.0969(1)	0.8606(1)	0.1195(7)	1
S2	2i	0.0713(1)	0.1765(1)	0.1658(2)	1
S3	2i	0.4771(3)	0.1600(5)	0.0320(8)	1
S4	2i	0.2227(9)	0.3824(6)	0.0471(5)	1
S5	2i	0.4427(5)	0.6694(1)	0.2116(4)	1

S6	2i	0.4119(8)	0.4914(6)	0.3493(3)	1
S7	2i	0.7905(8)	0.4906(2)	0.1926(6)	1
S8	2i	0.1708(8)	0.8406(1)	0.4033(8)	1
S9	2i	0.2455(7)	0.1721(4)	0.4538(1)	1
S10	2i	0.1655(5)	0.6781(1)	0.5640(8)	1
S11	2i	0.6719(2)	0.0228(8)	0.3124(3)	1
P1	2i	0.1728(9)	0.2062(4)	0.0346(3)	1
P2	2i	0.4691(3)	0.5071(9)	0.2034(4)	1
P3	2i	0.8902(7)	0.1600(4)	0.4415(6)	1
Li1	2i	0.1988(1)	0.3310(1)	0.4080(4)	1
Li2	2i	0.0401(1)	0.6381(6)	0.3657(2)	1
Li3	2i	0.7460(9)	0.2160(3)	0.2400(1)	1
Li4	2i	0.2720(9)	0.0580(8)	0.2300(1)	1
Li5	2i	0.0501(6)	0.6410(6)	0.1632(4)	1
Li6	2i	0.6290(5)	0.3630(6)	0.0020(6)	1
Li7	2i	0.4330(2)	0.1390(2)	0.6340(5)	1

Table S2-9. Crystallographic data of LPS w/o Al₂S₃. The lattice parameters, fractional atomic coordinates, occupancies are obtained from the Rietveld refinements against the XRD data.

Li ₃ PS ₄ (space group Pnm)					
$\lambda = 1.54051 \text{ \AA}$ lattice parameter $a = 12.8930(3) \text{ \AA}$, $b = 8.1362(6) \text{ \AA}$, $c = 6.1368(9) \text{ \AA}$					
$R_{wp} = 4.144 \%$, Goodness of fit = 2.04					
Phase fraction : 51.3 wt%					
Atom	Wyckoff position	Atomic coordinates			Occ.
		x	y	z	
S1	8d	0.1524(5)	0.0452(3)	0.2236(6)	1
S2	4c	0.4307(1)	0.25	0.2620(9)	1
S3	4c	0.0992(8)	0.25	0.6701(7)	1
P	4c	0.0788(8)	0.25	0.3169(1)	1
Li1	8d	0.3058(4)	0.0189(2)	0.1180(7)	1
Li2	4b	0	0	0.5	0.75
Li3	4c	0.0421(1)	0.25	0.4632(8)	0.25

Li ₇ P ₃ S ₁₁ (space group P-1)					
$\lambda = 1.54051 \text{ \AA}$ lattice parameter $a = 6.0513(5) \text{ \AA}$, $b = 12.3669(5) \text{ \AA}$, $c = 12.5398(5) \text{ \AA}$					
$R_{wp} = 4.144 \%$, Goodness of fit = 2.04					
Phase fraction : 48.7 wt%					
Atom	Wyckoff position	Atomic coordinates			Occ.
		x	y	z	
S1	2i	0.2156(5)	0.8660(5)	0.1050(5)	1
S2	2i	0.1259(8)	0.1571(1)	0.1501(8)	1
S3	2i	0.6742(2)	0.2071(1)	0.0340(4)	1

S4	2i	0.2937(1)	0.3895(6)	0.0998(1)	1
S5	2i	0.5493(5)	0.6920(8)	0.2234(1)	1
S6	2i	0.4850(7)	0.4845(5)	0.3428(2)	1
S7	2i	0.8865(3)	0.5079(8)	0.1995(6)	1
S8	2i	0.1795(7)	0.8347(9)	0.3936(1)	1
S9	2i	0.3159(8)	0.1838(2)	0.4488(1)	1
S10	2i	0.1754(4)	0.6771(1)	0.5969(9)	1
S11	2i	0.6668(1)	0.0379(2)	0.3465(7)	1
P1	2i	0.1944(5)	0.2131(6)	0.0071(4)	1
P2	2i	0.6374(3)	0.5545(7)	0.2064(6)	1
P3	2i	0.0204(2)	0.1523(8)	0.4291(1)	1
Li1	2i	0.0947(4)	0.4278(2)	0.4895(1)	1
Li2	2i	0.4713(3)	0.6088(6)	0.2939(1)	1
Li3	2i	0.4202(4)	-0.1153(7)	0.2817(1)	1
Li4	2i	-0.6459(1)	-0.1889(8)	-0.3307(6)	1
Li5	2i	0.6983(9)	0.6907(1)	-0.0227(8)	1
Li6	2i	0.6222(1)	0.4268(3)	-0.0761(1)	1
Li7	2i	0.9326(2)	0.4973(8)	0.5359(3)	1

Comment 3 The authors say that "a 2D ^{31}P refocused INADEQUATE experiment was performed for LPS7228". I can not understand this sentence. Please rephrase.

Response: To make this sentence more explicit, we have rephrased the wording of lines 140-142 to: "a 2D ^{31}P - ^{31}P INADEQUATE NMR experiment was performed for LPS7228. The 2D experiment is based on J -coupling which allows us to probe P-P and/or P-S-P bond connectivity."

Changes in the manuscript: "a 2D ^{31}P - ^{31}P INADEQUATE NMR experiment was performed for LPS7228. The 2D experiment is based on J -coupling which allows us to probe P-P and/or P-S-P bond connectivity".

Comment 4 When the authors describe the compositions of the glass-ceramics, the sum of mass concentrations is higher than 100%. While reading it becomes more clear what do the authors mean. However, I suggest more straightforward description of the compositions through all the manuscript and supplementary info.

Response: Thanks for your comment. In order not to confuse the readers, we have re-normalized the stoichiometry of raw materials in Table 1.

Table 1. Molar ratio of the reagents (Li_2S , P_2S_5 , Al_2S_3) in Al-GCs and the corresponding denotation.

Li ₂ S	P ₂ S ₅	Al ₂ S ₃	Ratio of (Li ₂ S : P ₂ S ₅)	Denotation
56.25	18.75	0	75 : 25	LPS7525
55.5	19.5	1	74 : 26	LPS7426
54.75	20.25	2	73 : 27	LPS7327
54	21	3	72 : 28	LPS7228
53.25	21.75	4	71 : 29	LPS7129
52.5	22.5	5	70 : 30	LPS7030
51.75	23.25	6	69 : 31	LPS6931
48.75	26.25	10	65 : 35	LPS6535

REVIEWERS' COMMENTS

Reviewer #1 (Remarks to the Author):

All my questions have been clearly answered, I recommend publication of this manuscript in Nature Communications.

Reviewer #2 (Remarks to the Author):

The resulting self-organized hetero-nanodomains, composed of a considerable number of grain boundaries, can actuate super Li⁺ conduction, leading to a high room temperature of 13.2 mS cm⁻¹ in the cold-pressed state. Also, high energy density all-solid stated sulfide-based batteries had been achieved.

Yes. This work is of great significance to the field and the related fields and presents a new approach to increasing Li⁺ conductivity compared to the literature.

Yes. This work is original.

Yes. The work supports the conclusions and claims.

No. There are no flaws in the data analysis, interpretation, or conclusions.

Yes. The methodology is sound. The work meets the expected standards in our field.

Yes. The provided details in the methods are enough for the work to be reproduced.

Reviewer #3 (Remarks to the Author):

I've thoroughly analysed the changes made by the authors in response to the reviewers' comments and the authors' answers. In my opinion, the manuscript can be now accepted for publication.

Responses to the reviewers' comments

Reviewer #1:

All my questions have been clearly answered, I recommend publication of this manuscript in Nature Communications..

Response: We thank the reviewer for very valuable comments in the first round of peer review.

Reviewer #2:

General Comments: *The resulting self-organized hetero-nanodomains, composed of a considerable number of grain boundaries, can actuate super Li⁺ conduction, leading to a high room temperature of 13.2 mS cm⁻¹ in the cold-pressed state. Also, high energy density all-solid stated sulfide-based batteries had been achieved.*

Yes. This work is of great significance to the field and the related fields and presents a new approach to increasing Li⁺ conductivity compared to the literature.

Yes. This work is original.

Yes. The work supports the conclusions and claims.

No. There are no flaws in the data analysis, interpretation, or conclusions.

Yes. The methodology is sound. The work meets the expected standards in our field.

Yes. The provided details in the methods are enough for the work to be reproduced.

Response: We thank the reviewer for the strong endorsement of our work.

Reviewer #3:

General Comments: *I've thoroughly analysed the changes made by the authors in response to the reviewers' comments and the authors' answers. In my opinion, the manuscript can be now accepted for publication.*

Response: We thank the reviewer for this positive feedback and for all their comments that were instrumental in improving the overall quality of our work.